# SLC6A14, an amino acid transporter, modifies the primary CF defect in fluid secretion

**Saumel Ahmadi[1,2], Sunny Xia[1,2], Yu-Sheng Wu[1,2], Michelle Di Paola[2,3], Randolph Kissoon[2], Catherine Luk[2], Fan Lin[4], Kai Du[2], Johanna Rommens[4,5], Christine E Bear[1,2,3]***

[1]Department of Physiology, University of Toronto, Toronto, Canada; [2]Programme in Molecular Medicine, Research Institute, Hospital for Sick Children, Toronto, Canada; [3]Department of Biochemistry, University of Toronto, Toronto, Canada; [4]Department of Molecular Genetics, University of Toronto, Toronto, Canada; [5]Programme in Genetics and Genome Biology, Research Institute, Hospital for Sick Children, Toronto, Canada

**Abstract** The severity of intestinal disease associated with Cystic Fibrosis (CF) is variable in the patient population and this variability is partially conferred by the influence of modifier genes. Genome-wide association studies have identified *SLC6A14,* an electrogenic amino acid transporter, as a genetic modifier of CF-associated meconium ileus. The purpose of the current work was to determine the biological role of *Slc6a14,* by disrupting its expression in CF mice bearing the major mutation, F508del. We found that disruption of *Slc6a14* worsened the intestinal fluid secretion defect, characteristic of these mice. In vitro studies of mouse intestinal organoids revealed that exacerbation of the primary defect was associated with reduced arginine uptake across the apical membrane, with aberrant nitric oxide and cyclic GMP-mediated regulation of the major CF-causing mutant protein. Together, these studies highlight the role of this apical transporter in modifying cellular nitric oxide levels, residual function of the major CF mutant and potentially, its promise as a therapeutic target.
DOI: https://doi.org/10.7554/eLife.37963.001

**\*For correspondence:**
bear@sickkids.ca

**Competing interests:** The authors declare that no competing interests exist.

## Introduction

Cystic Fibrosis (CF) is the most common fatal genetic disorder, caused by mutations in the Cystic Fibrosis Transmembrane Conductance Regulator (*CFTR*) gene (*Kartner et al., 1991*; *Li et al., 1993*; *Riordan et al., 1989*; *Rommens et al., 1989*). The CFTR protein is primarily expressed in epithelial cells lining the airways, intestinal tract, and tubular organs, where its anion channel activity on the apical membrane drives fluid transport and modulates the pH of elaborated fluid (*Bear et al., 1992*; *Li et al., 1993*; *Quinton, 1990*; *Shah et al., 2016a*, *2016b*). It is the loss of this channel function, caused by *CFTR* mutations, that triggers pathogenesis. However, the severity of disease amongst individuals harboring the same genetic mutation is variable (*Kerem et al., 1990*; *Kerem and Kerem, 1996*; *Luisetti, 1997*; *Rosenstein, 1994*).

Decrease in lung function over time is the most common cause of morbidity and mortality in CF patients (*Gilljam et al., 2004*; *Kerem and Kerem, 1996*) and recent genome-wide association studies have identified polymorphisms in several secondary genetic factors associate with CF lung disease severity (*Corvol et al., 2015*; *Sun et al., 2012*). CF patients also exhibit gastrointestinal disease manifestations, such as meconium ileus (MI) at birth, and distal intestinal obstructive syndrome (DIOS) (*Canale-Zambrano et al., 2007*; *Werlin et al., 2010*). The intestinal phenotype of MI can be

easily diagnosed in neonates at birth, and is highly heritable (>88%), having minimal environmental influence (*Blackman et al., 2006*). For this reason, it was used in the genome-wide association studies (GWAS), which identified *SLC6A14*, *SLC26A9* and *SLC9A3* as modifiers of the CF intestinal phenotype (*Sun et al., 2012*).

The role for secondary genes in modifying CF disease severity has been studied extensively using CF mouse models (*Bradford et al., 2009*; *Hillesheim et al., 2007*; *Liu et al., 2015*; *Rozmahel et al., 1996*; *Singh et al., 2013*; *Walker et al., 2008*). Deletion of the *Cftr* gene, or knock-in of the mutant F508del *Cftr* gene, generates significant changes to intestinal pathology (*Grubb and Gabriel, 1997*; *Ratcliff et al., 1993*; *Scholte et al., 2004*; *van Doorninck et al., 1995*). CF mice have growth retardation when compared to their Wt (wild type) littermates, and this has been attributed to malabsorption and decreased secretion of IGF-1 (*Canale-Zambrano et al., 2007*; *Rogan et al., 2010*; *van Doorninck et al., 1995*). Histologically, the intestine of CF mice exhibits mucus accumulation, inflammation and goblet cell hyperplasia in the epithelial layers (*Grubb and Gabriel, 1997*; *Ratcliff et al., 1993*), and circular smooth muscle hypertrophy in the muscularis externa (*Risse et al., 2012*). This increase in smooth muscle thickness of the intestinal wall is variable in CF mice of different backgrounds (*Bazett et al., 2015*; *Risse et al., 2012*), and modifier genes have been attributed to these differences. The role of *Slc26a9* and *Slc9a3* in modifying the CF phenotype has been examined by disrupting the expression of these genes in CF mouse models. Disruption of *Slc26a9* caused defects in bicarbonate secretion and fluid absorption in the proximal duodenum, leading to increased mortality in CF mice (*Liu et al., 2014*). On the other hand, disruption of *Slc9a3* expression improved the CF phenotype of fluid secretion and reversed the intestinal phenotype of CF mice (*Bradford et al., 2009*).

SLC6A14 is a $Na^+/Cl^-$ dependent neutral and cationic amino acid transporter (*Karunakaran et al., 2011*; *Rajan et al., 2000*) expressed on the apical membrane of epithelial cells. It is hypothesized that this amino acid transporter is principally involved in nutrient uptake, due to its broad specificity and concentrative transport mechanisms (*Galietta et al., 1998*; *Rudnick et al., 2014*) Furthermore, it has been studied as a potential drug target in various epithelial cancers, such as colon, breast and pancreats (*Babu et al., 2015*; *Coothankandaswamy et al., 2016*; *Karunakaran et al., 2011*; *Karunakaran et al., 2008*). However, to date, the biological role of SLC6A14 in modifying the CF phenotype has not been interrogated. The aim of the current study is to determine the impact of disrupting *Slc6a14* expression in CF mice harbouring the major CF causing mutation: F508del.

## Results

### *Slc6a14* is a major apical amino acid transporter in the colon

Quantification of relative mRNA expression by qRT-PCR revealed that *Slc6a14* is expressed predominantly in the wild-type mouse colon (C57BL/6N *Figure 1a*). In order to define SLC6A14 protein localization in colonic epithelium, we transduced mouse colonic organoids with lentivirus containing *SLC6A14-GFP,* or *GFP* alone as a control, and examined localization by confocal microscopy. We found that SLC6A14 was localized at the apical pole on the apical surface, as expected (*Figure 1b*). The subsequent series of experiments were designed to interrogate the biological role of SLC6A14 in mediating amino acid transport in the mouse colon, using mice in which *Slc6a14* expression was abolished. First, we confirmed that the *Slc6a14* gene was disrupted, and expression was abrogated in the knock-out (KO) mouse (C57BL/6N), created by the NORCOMM Consortium (*Figure 1c,d*). *Slc6a14* is located on the X chromosome, hence the generation of null male mice was more frequent, and our studies focused on this gender.

There are many known apically expressed amino acid transporters in the colonic epithelium (*Bröer, 2008*; *Sloan and Mager, 1999*; *Ugawa et al., 2001*), hence, we asked if deletion of *Slc6a14* impacts amino acid uptake across the apical membrane of this epithelium. To address this, we measured $^3$H-arginine uptake across the apical surface of the colonic epithelium, since arginine is an important substrate for respiratory health (*Coburn et al., 2012*; *Grasemann et al., 2011*; *Grasemann and Ratjen, 2012*; *Grasemann et al., 2005*). We used the ex vivo closed loop assay to measure arginine uptake in these mice (*Ugawa et al., 2001*). We performed this assay on the colon from C57BL/6N wild-type and *Slc6a14* $^{(-/y)}$ mice. Interestingly, we found that the *Slc6a14* $^{(-/y)}$ mice exhibited ~ 75% reduction in apical arginine transport (*Figure 1e*), suggesting that *Slc6a14*

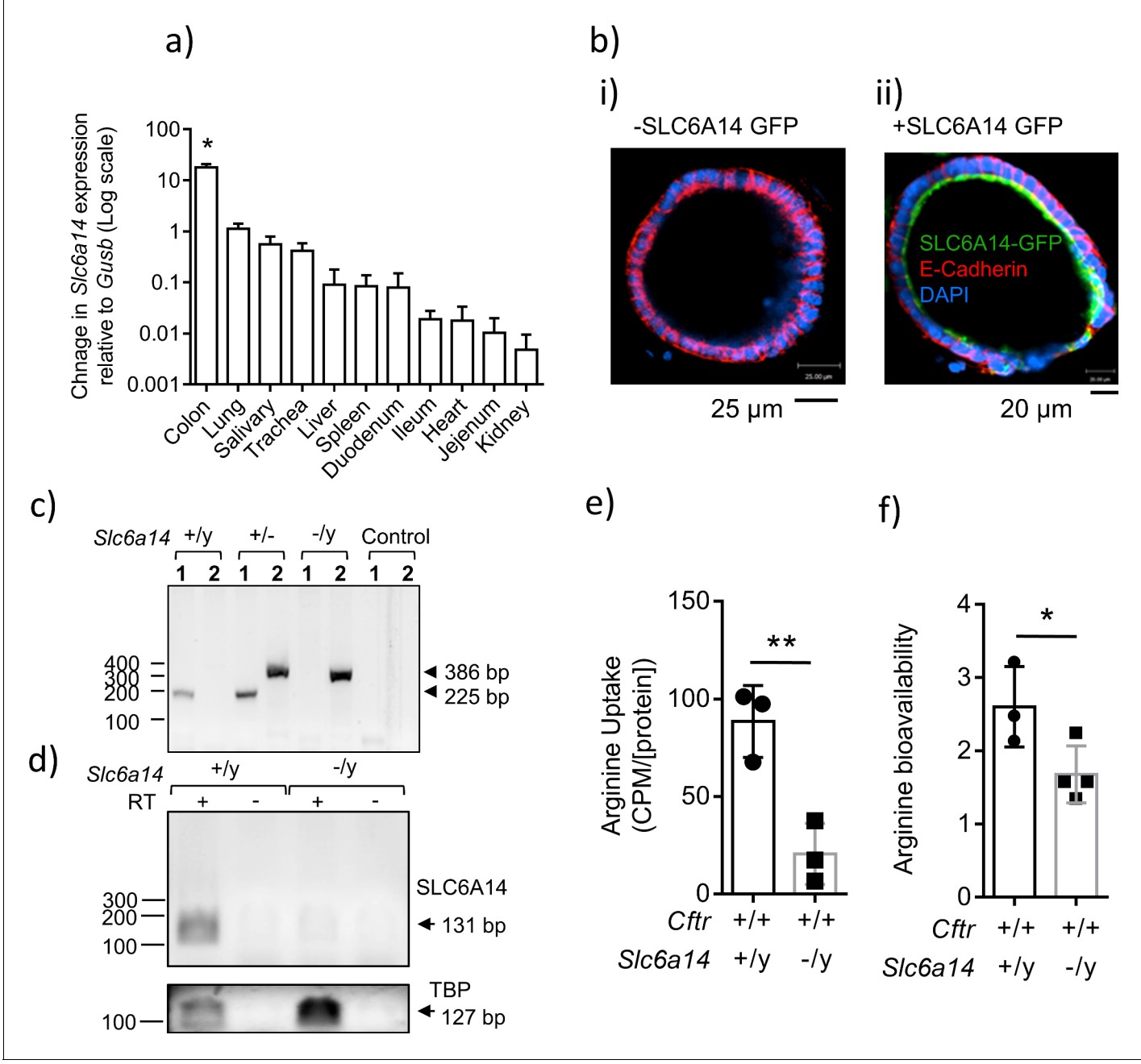

**Figure 1.** Expression of *SLC6A14* in various CF affected tissues. (**a**) mRNA Expression of *Slc6a14* normalized to housekeeping gene *Gusb* was measured in various CF-affected tissues, using qRT-PCR. Bars represent mean ± SEM. One-way ANOVA with Tukey's multiple comparison test was performed (*p<0.0001, n ≥ 3 biological replicates). (**b**) Colonic organoids derived from a wild-type C57BL/6N mice were transduced with human *SLC6A14-GFP*, fixed and immunostained for GFP, murine E-Cadherin and DAPI (i) Immunofluorescence of non-transduced colonic organoids. (ii) Fluorescence imaging showing apical localization of *SLC6A14-GFP*. (**c**) For genotype confirmation, DNA was collected from *Slc6a14*[(+/y)] and *Slc6a14*[(-/y)] mice tails and PCR was performed (35 cycles). Two different sets of primers were used (set 1, 2). Set one amplified the wild-type allele only (225 bp), and primer set two amplified the *Slc6a14* knock-out allele only (386 bp). Hence, two bands are observed in heterozygous mice. (**d**) *Slc6a14* mRNA expression in the colon of Wt and *Slc6a14*-KO mice was measured by PCR of the cDNA (30 cycles). *Tbp* was used as the housekeeping gene. (**e**) Ex vivo closed loop assay was performed after injecting buffer containing [3]H-arginine supplemented with 100 µM or 20 mM cold arginine. After 15 mins, the epithelium was lysed and intracellular [3]H-arginine levels measured. Bar graph represents mean ± SD of arginine uptake by the epithelium (counts per minute, CPM), normalized to total protein in the lysate (CPM/[protein]). Unpaired t-test was performed (**p=0.0082; n = 3 biological replicates). (**f**) Bar graphs represent arginine bioavailability in freshly lysed colonic tissue (mean ± SD). Arginine bioavailability is defined as the ratio of arginine to citrulline plus ornithine. Unpaired t-test was performed (*p=0.047; n = 3 biological replicates).

DOI: https://doi.org/10.7554/eLife.37963.002

*Figure 1 continued on next page*

*Figure 1 continued*

The following figure supplement is available for figure 1:

**Figure supplement 1.** Deletion of *Slc6a14* in CF mice does not cause a change in serum amino-acid levels.

DOI: https://doi.org/10.7554/eLife.37963.003

constitutes a major apical arginine uptake pathway in the mouse colon. Interestingly, decrease in amino acid uptake across the colonic epithelium did not affect serum amino acid levels (*Figure 1—figure supplement 1*, *Supplementary file 1*).

Since SLC6A14 played a major role in apical arginine flux, we asked if the steady state levels of arginine are affected in the colonic epithelium of the *Slc6a14* knock-out mouse. We performed mass spectrometry using freshly lysed tissue, and found that the steady state levels of arginine were indeed lower in the *Slc6a14* knock-out colonic epithelium (steady state L-arginine = L-arginine/(L-citrulline + L-ornithine)), compared to wild-type epithelium (*Figure 1f*). Thus, we found that SLC6A14 is a major apically expressed amino acid transporter in the colon, and it helps maintain the steady state levels of arginine in the colonic epithelium.

## Disruption of *Slc6a14* retards weight gain of mice modeling the major CF mutation: F508del, at weaning

To study the impact of *Slc6a14* knock-out on the CF affected epithelium, we generated a double mutant mouse. It is known that the F508del mutation in mice results in an intestinal phenotype of varying severity, which is dependent on the genetic background of the mouse (*Rozmahel et al., 1996*). We hypothesized that *Slc6a14* knock-out would worsen the CF phenotype, hence, we chose the *Cftr*$^{(F508del/F508del)}$ mouse, using the FVB strain, as it exhibits a relatively mild CF intestinal phenotype due to residual F508del-CFTR function and would permit quantification of a deleterious effect (*van Doorninck et al., 1995*). We backcrossed the C57BL/6N *Slc6a14* knock-out mouse to the FVB *Cftr* F508del heterozygous mouse, for >7 generations, to generate a congenic mouse strain carrying a double mutation of *Cftr*$^{(F508del/F508del)}$ *Slc6a14*$^{(-/y)}$. We confirmed a Mendelian inheritance for the two genes in this congenic strain (*Supplementary file 2*, *Figure 2* ).

It has been previously reported that CF mice exhibit reduced weights relative to their wild-type siblings early in life after weaning (*van Doorninck et al., 1995*), and we confirmed this defect in the current experiment at day 34 after birth, in the *Cftr*$^{(F508del/F508del)}$ mice (*Figure 2*). As shown in *Figure 2*, disruption of *Slc6a14* in CF mice (*Cftr*$^{(F508del/F508del)}$ *Slc6a14*$^{(-/y)}$) led to a further reduction in weight gain and BMI post weaning (day 34), relative to age matched controls, (*Cftr*$^{(F508del/F508del)}$ *Slc6a14*$^{(+/y)}$) (*Figure 2a,b,c*, *Figure 2—figure supplement 1*). While this defect in weight gain was temporary in nature, with the double mutant mice recovering to the same weight as the F508del mice by 56 days in age, it was associated with increased mortality during this period (*Figure 2d*).

The basis for weight reduction in *Cftr*$^{(F508del/F508del)}$ mice is not fully understood but is thought to be related to decreased food intake (although there is some controversy) (*De Lisle and Borowitz, 2013*; *Rosenberg et al., 2006*), reduced IGF-1 levels (*Rosenberg et al., 2006*), and/or changes in the tissue secondary to dehydrated and static mucus (*De Lisle and Borowitz, 2013*; *Scholte et al., 2004*; *van Doorninck et al., 1995*), hallmark features of Cystic Fibrosis. Hence, we reason that one or all of these factors may be worsened in the double mutant mice although, in contrast to previous studies, we did not observe a sustained decrease in IGF-1 (*Figure 2—figure supplement 2*)

## Loss of *Slc6a14* worsens defective F508del-*Cftr*-mediated secretion in murine colonic epithelium

In order to assess the consequences of *Slc6a14* knock-out on the intestinal tissue morphology of CF mice, we conducted histological analyses of the small and large intestines as described in the Materials and methods and *Figure 3—figure supplement 1*. Sections from the large intestine confirmed previous reports of mucus accumulation and increased smooth muscle thickness in the ileum and colon of the F508del mice relative to wild-type mice (*Canale-Zambrano et al., 2007*; *De Lisle et al., 2012*; *Risse et al., 2012*; *Rosenberg et al., 2006*). Interestingly, we also found that the CF-related smooth muscle thickening phenotype in the distal ileum and colon was worse when *Slc6a14* was disrupted (*Figure 3a,b,c and d*).

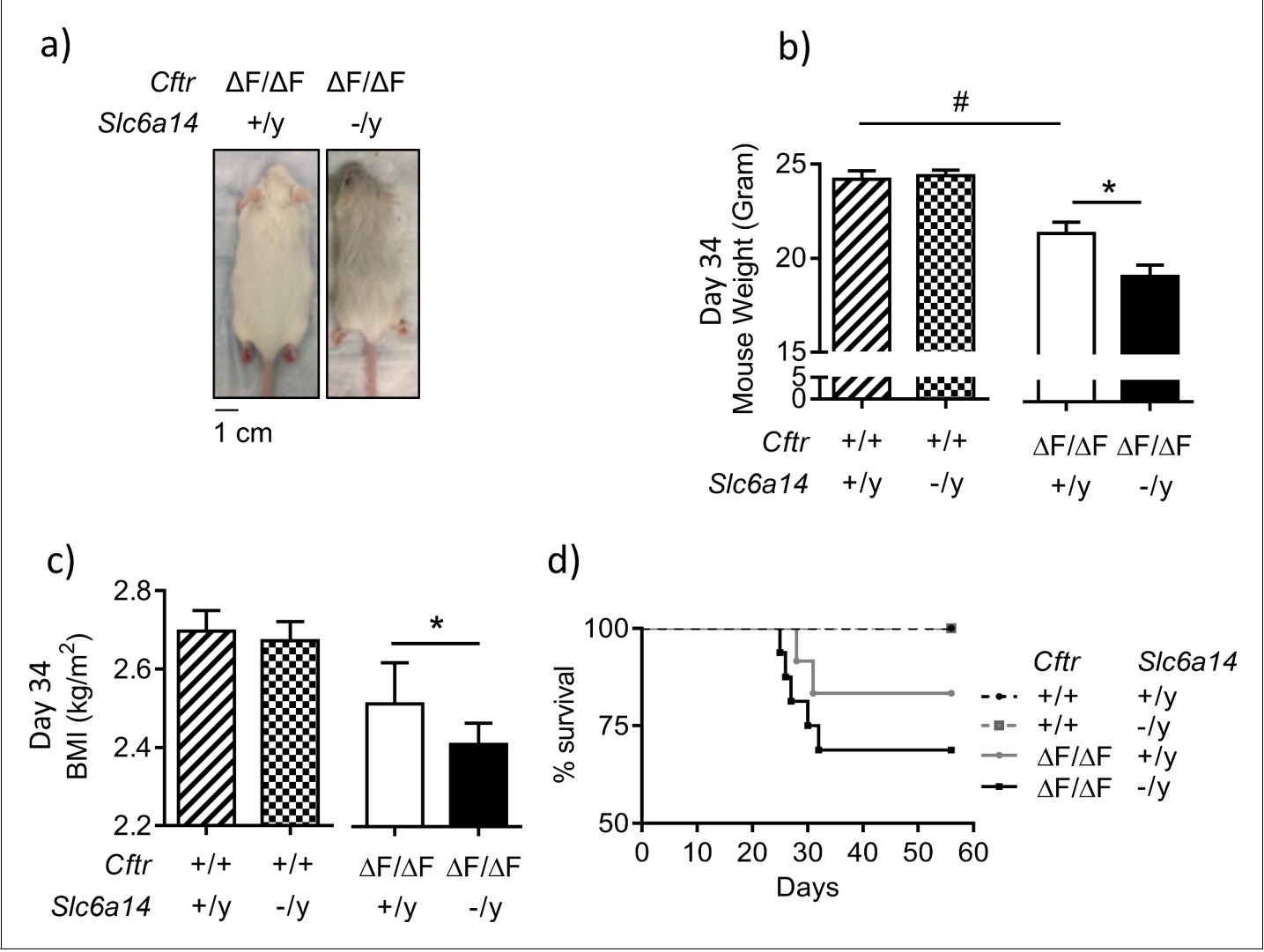

**Figure 2.** Disruption of *Slc6a14* in F508del CF mice leads to decrease in weight gain post weaning. (a) Dorsal view of CF (*Cftr*[F508del/F508del]) mice and double mutant (*Cftr*[F508del/F508del]; *Slc6a14*[(-/y)]) mice. Both were day 30 male mice. (b) Bar graph represents weights of Wt, *Cftr*[F508del/F508del], *Slc6a14*[(-/y)] and double mutant (*Cftr*[F508del/F508del]; *Slc6a14*[(-/y)]) mice at day 34 (mean ± SEM). Unpaired t-test was performed (#p=0.005, *p=0.04; n ≥ 4 mice for each genotype). (c) Bar graph represents Body Mass Index (BMI) of mice at day 34 (mean ± SEM). Unpaired t-test was performed (*p=0.03; n > 5 mice for each genotype). (d) The graph represents Kaplan-Meier survival curves for the four genotypes of mice. Double mutant mice were susceptible to death post-weaning. There was no statistical difference in the survival, between Wt and F508del-*Cftr* mice, However, the survival of double mutant mice was significantly lower than *Slc6a14*[(-/y)] mice (Log Rank test, p=0.0036; Wt mice n = 14, *Slc6a14*[(-/y)]n = 15, *Cftr*[F508del/F508del]n = 10, double mutant - *Cftr*[F508del/F508del]; *Slc6a14*[(-/y)]n = 11).

DOI: https://doi.org/10.7554/eLife.37963.004

The following figure supplements are available for figure 2:

**Figure supplement 1.** Line graph represents weight gain over time in Wt, *Cftr*[F508del/F508del], *Slc6a14*[(-/y)] and double mutant (*Cftr*[F508del/F508del]; *Slc6a14*[(-/y)]) mice.

DOI: https://doi.org/10.7554/eLife.37963.005

**Figure supplement 2.** Deletion of *Slc6a14* in CF mice does not cause a change in serum IGF-1 levels.

DOI: https://doi.org/10.7554/eLife.37963.006

We hypothesized that the worsening of the smooth muscle thickness phenotype in the double mutant mice is secondary to an exacerbated defect in the epithelium. We employed the in-situ, closed loop assay described by Verkman and colleagues (*Sonawane et al., 2006*) to examine fluid secretion by the epithelium. As previously reported, cyclic AMP (cAMP) induced CFTR-mediated fluid accumulation was decreased in intestinal segments derived from mice, homozygous for

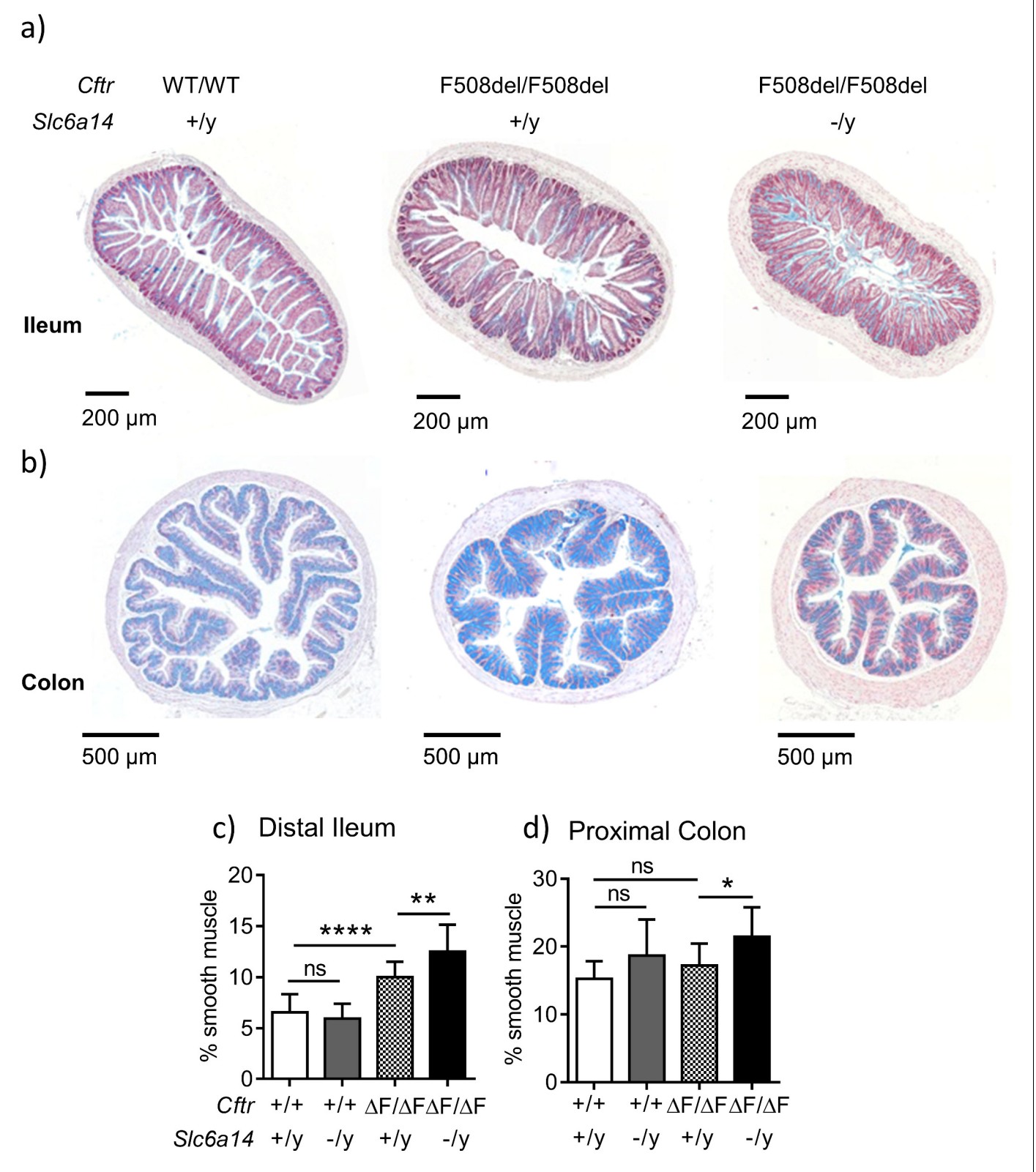

**Figure 3.** *Slc6a14* disruption results in an increase in smooth muscle thickness in the ileum and colon. (**a**) Representative distal ileal sections stained with Alcian blue, for Wt, *Cftr*F508del/F508del and double mutant (*Cftr*F508del/F508del; *Slc6a14*(-/y)) mice. (**b**) Representative colonic sections stained with Alcian blue, for Wt, *Cftr*F508del/F508del and double mutant (*Cftr*F508del/F508del; *Slc6a14*(-/y)) mice. (**c**) Bar graph represents ileal smooth muscle area relative to total section area for each genotype (mean ± SD). One-Way ANOVA with Tukey's multiple comparison test was performed (**p=0.007, ****p<0.0001, n ≥ 4

*Figure 3 continued on next page*

*Figure 3 continued*

mice for each genotype). (**d**) Bar graph represents colon smooth muscle area relative to total section area for each genotype (mean ± SD). One-Way ANOVA with Tukey's multiple comparison test was performed (*p=0.042, n ≥ 4 mice for each genotype).

DOI: https://doi.org/10.7554/eLife.37963.007

The following figure supplement is available for figure 3:

**Figure supplement 1.** Representative intestinal section with annotations (green line) made in 3DHISTECH software are shown. These annotations were used to calculate relevant areas of tissue layers.

DOI: https://doi.org/10.7554/eLife.37963.008

F508del-*Cftr* (*Figure 4a and b*). Importantly, disruption of *Slc6a14* in the CF mice, further worsened this secretory defect (*Figure 4b*). Interestingly, *Slc6a14* disruption alone impaired cAMP-induced secretion relative to that in wild-type (C57BL/6N background) siblings (*Figure 4—figure supplement 1*). This decrease in fluid secretion was not due to change in *Cftr* mRNA expression (*Figure 4—figure supplement 1*), and the basis for this change is investigated in subsequent studies.

We further interrogated the role of SLC6A14 in fluid secretion using colonic organoids (*Dekkers et al., 2013*; *Sato et al., 2009*), an adult stem cell-derived model that reports CFTR-mediated fluid secretion (*Foulke-Abel et al., 2016*; *Saxena et al., 2016*). We generated colonic organoids from double mutant and F508del-CF mice, and studied residual CFTR function in these organoids using the forskolin-induced swelling (FIS) assay (*Dekkers et al., 2013*) as described in detail in the Materials and methods and *Figure 4—figure supplement 2*. As expected from the above 'closed loop' studies, the FIS response was significantly lower in the colonic organoids from double mutant mice ($Cftr^{F508del/F508del}$; $Slc6a14^{(-/y)}$), compared to the F508del-CF mice (*Figure 4c and d*). Taken together, the results of the in-situ closed loop assay and the FIS assay show that SLC6A14 modifies F508del-CFTR-mediated secretion.

## Loss of *Slc6a14* and arginine-mediated nitric oxide generation contributes to worsening of defective epithelial fluid secretion

The above organoid swelling studies suggest that the function of SLC6A14 enhances the residual channel function of murine F508del-CFTR. We confirmed that SLC6A14 is a major apical arginine transporter in the colonic epithelium of wild-type and F508del mice, on the FVB background (*Figure 5—figure supplement 1*). Hence, we hypothesized that the apical uptake of arginine via SLC6A14 augments residual cAMP-dependent chloride channel activity by F508del-CFTR (*Figure 5a*). In order to test this directly, we 'opened' the organoid structures in order to provide access to the apical membrane of the intestinal epithelium, and employed a fluorescence-based method to study changes in the apical membrane chloride conductance associated with CFTR channel opening. While the Ussing chamber technique is typically used to measure apical chloride channel activity in intestinal tissue, this proved to be problematic in studies of tissue from double mutant mice ($Cftr^{F508del/F508del}$; $Slc6a14^{(-/y)}$), given the small size of the colon of these animals.

The split-open colonic organoid resembles a round lawn of cells with the apical surface exposed to the bath solution and shown in *Figure 5—figure supplement 2*. This configuration enables direct assessment of apical F508del-CFTR channel function. F508del-CFTR-mediated chloride conductance across the apical membrane can be measured using a fluorescence-based assay (the ACC assay), previously optimized for the study of clinical samples (*Ahmadi et al., 2017*; *Molinski et al., 2015*). Colonic organoids from F508del mice were split open and CFTR channel function was measured using the ACC assay (*Figure 5c,d*). The ACC assay was initially conducted at 27°C, an experimental maneuver, well known to enhance the trafficking of F508del to the cell surface (*Denning et al., 1992*). Importantly, we found that acute pre-addition of L-arginine (1 mM) on the apical surface, increased the residual, cAMP-activated function of F508del-CFTR in split-open organoids derived from F508del-mice at 27°C (*Figure 6a and b*). This arginine-dependent enhancement effect was lacking in tissues obtained from the double mutant mice, lacking SLC6A14 (*Figure 5a and b*).

Since we could detect a significant response to the cAMP agonist in split open organoids obtained from F508del-CFTR mice and its modulation by arginine at 27°C, we were prompted to re-examine this response at the physiologically relevant temperature of 37°C, a temperature expected to reduce F508del-CFTR function at the cell surface. F508del-CFTR channel function upon FSK

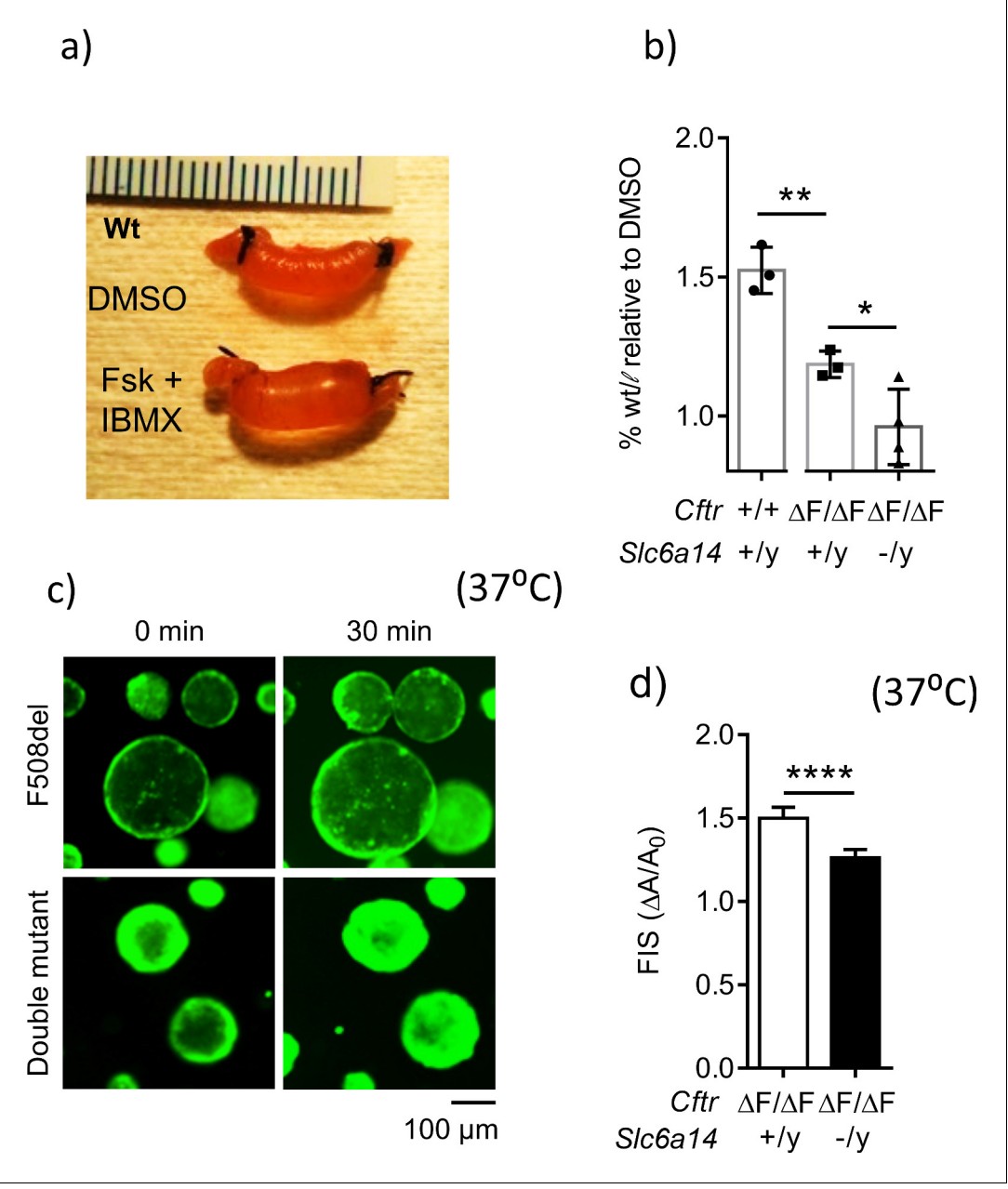

**Figure 4.** Loss of *Slc6a14* worsens defective F508del *Cftr*-mediated secretion in murine colonic epithelium. (a) In vivo closed loop assay performed on Wt mice. Each loop was injected with CFTR cAMP agonist forskolin (FSK 10 μM) and IBMX (100 μM), or DMSO vehicle. Weight relative to length is used as a measure of fluid secretion. (b) Bar graph represents fluid secretion in loops stimulated with CFTR cAMP agonist FSK and IBMX, relative to DMSO alone (mean ± SD). Fluid secretion is represented as weight/length for each loop. Wt mice showed significantly higher fluid secretion than $Cftr^{F508del/F508del}$ (unpaired t-test, # p=0.0036, n = 3 for each genotype). Double mutant ($Cftr^{F508del/F508del}$; $Slc6a14^{(-/y)}$) mice showed significantly lower fluid secretion than CF $Cftr^{F508del/F508del}$ mice (unpaired t-test, *p=0.0428, **p=0.0036, n ≥ 3 biological replicates for each genotype). (c) Representative fluorescence images of murine colonic organoids derived from $Cftr^{F508del/F508del}$ and double mutant ($Cftr^{F508del/F508del}$; $Slc6a14^{(-/y)}$) mice, and their responses to 30 min of stimulation with FSK (1 μM). (d) Bar graph represents FSK- induced swelling (FIS) after 30 min of stimulation, in both $Cftr^{F508del/F508del}$ and double mutant ($Cftr^{F508del/F508del}$; $Slc6a14^{(-/y)}$) murine organoids (mean ± SD). FIS is measured as change in area of the organoid after 30 min of FIS (ΔA) relative to baseline before stimulation ($A_0$). Unpaired t-test was performed (****p<0.0001, n > 4 biological replicates for each genotype).

DOI: https://doi.org/10.7554/eLife.37963.009

*Figure 4 continued on next page*

*Figure 4 continued*

The following figure supplements are available for figure 4:

**Figure supplement 1.** Disruption of *Slc6a14* decreases the fluid secretory capacity of the colonic epithelium by modulating apical constituents of the epithelium.

DOI: https://doi.org/10.7554/eLife.37963.010

**Figure supplement 2.** Analysis of FSK-induced swelling of murine colonic organoids.

DOI: https://doi.org/10.7554/eLife.37963.011

stimulation was still detectable at this temperature, albeit reduced in comparison to measurements acquired at 27°C (*Figure 5e,f*), reflecting the sensitivity of the ACC assay. Further, as in the preceding experiments, this residual mutant CFTR function, was enhanced by pre-addition of L-arginine on the apical surface in the $Cftr^{(F508del/F508del)}$ murine organoids, but not from those derived from the tissue of double mutant ($Cftr^{F508del/F508del}$; $Slc6a14^{(-/y)}$) mice (*Figure 5e,f*). Also, the cAMP-stimulated CFTR function was lower in the double mutant-derived organoids. Hence, SLC6A14-mediated regulation of murine F508del-CFTR occurs at the physiologically relevant temperature.

SLC6A14-mediated increase in F508del-CFTR function could be achieved through a number of possible mechanisms. The mutant F508del CFTR protein is known to have several defects at the cellular level, including retention in the ER (*Cheng et al., 1990*), and decreased surface stability of the temperature rescued protein (*Lukacs et al., 1993*). We tested the impact of SLC6A14 on these cellular phenotypes of human F508del CFTR protein, using the BHK heterologous over-expression system. We found that over-expression of SLC6A14 did not affect the processing of the temperature rescued F508del-CFTR protein (*Figure 5—figure supplement 3*), nor did it alter the cell surface stability of the rescued protein. Therefore, there was no significant effect of SLC6A14 expression on the processing or stability of human F508del-CFTR. These findings are consistent with the results of our biochemical studies, that failed to provide conclusive support for a direct or indirect interaction between these two membrane proteins. As shown in *Figure 5—figure supplement 4*, we did not detect reciprocal co-immunoprecipitation of the two membrane proteins. These negative results prompted us to test the hypothesis that the positive functional interaction between SLC6A14 and F508del-CFTR as described in *Figures 4* and *5* may be mediated via intracellular signaling.

As shown in *Figure 6a*, the SLC6A14 substrate, arginine, is converted to nitric oxide (NO) by nitric oxide synthase (NOS), and we found that induced NOS (iNOS) is expressed in mouse colonic tissue (*Figure 6—figure supplement 1*). Nitric oxide is known to exert a positive effect on the functional expression of Wt-CFTR (*Figure 6a*), (*Zaman et al., 2006*). In the current work, we showed nitric oxide (NO) also activates F508del-CFTR channel function. Specifically, the forskolin response measured in split-open organoids from $Cftr^{F508del/F508del}$ mice was enhanced by the NO donor GSNO (*Figure 6—figure supplement 2*). The role of SLC6A14 transported arginine and iNOS in modulating F508del-CFTR channel function was tested in comparative studies of organoids from $Cftr^{F508del/F508del-}$ versus $Cftr^{F508del/F508del}$; $Slc6a14^{(-/y)}$ mice. We found that the positive effect of arginine on forskolin activated F508del channel function was reduced by pretreatment with the iNOS inhibitor 1400W in organoids derived from $Cftr^{F508del/F508del}$ mice. Further the 1400W sensitive effect of arginine was dependent on SLC6A14 expression (*Figure 6b,c*). Given this initial evidence supporting a role for SLC6A14 mediated NO signaling in enhancing F508del-CFTR channel function, we were prompted to assess the role of SLC6A14 in NO synthesis.

In these studies, a fluorescence-based assay was used to measure intracellular NO levels (DAF-FM-DA), in murine intestinal tissue. A standard curve for change in DAF-FM fluorescence to increasing concentrations of NO was generated (*Figure 6—figure supplement 3*). As shown in *Figure 6d*, colonic epithelial tissue derived from *Slc6a14* knock-out mice exhibited lower constitutive NO levels. Interestingly, while there was an increase in NO production evoked by acute addition of arginine (1 mM) to the apical surface of split-open colonic organoids obtained from wild-type mice, this increase was abrogated in split-open organoids generated from $Slc6a14^{(-/y)}$ littermates (*Figure 6e–f*). Altogether, these data show that *Slc6a14*-mediated arginine uptake is important for NO synthesis, and hence NO mediated signaling to the CFTR channel. Conversely, over-expression of *SLC6A14* via lentivirus transduction conferred increased constitutive levels of NO and enhanced the increase in cytosolic NO after addition of arginine to the apical surface of split-open organoids (*Figure 6g–i*).

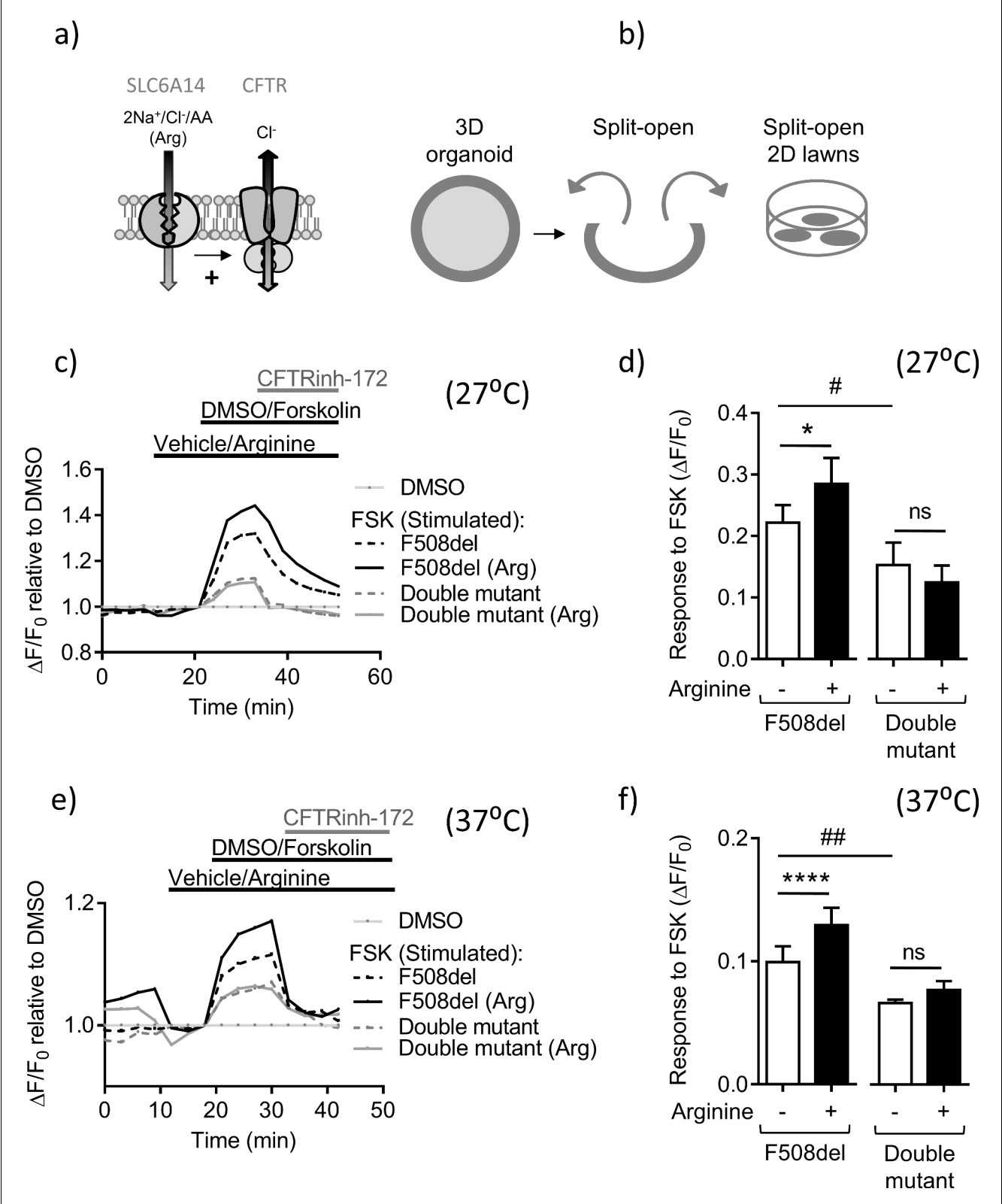

**Figure 5.** *Slc6a14* expression mediates arginine-dependent enhancement of mutant F508del CFTR channel function in murine intestinal tissues. (a) Hypothetical model depicting that SLC6A14 could affect CFTR channel function. (b) Diagram depicts the concept of gaining apical access to the epithelium by splitting open a 3D organoid, thereby resulting in patches of split-open 2D lawns, which can then be studied using fluorescence-based assays. (c) Split-open colonic organoids from CF (*Cftr*[F508del/F508del]) and double mutant (*Cftr*[F508del/F508del]; *Slc6a14*[(-/y)]) mice were studied for CFTR

*Figure 5 continued on next page*

*Figure 5 continued*

channel function using the previously described membrane potential-based ACC assay. Line graph represents change in fluorescence relative to baseline ($\Delta F/F_0$) as a measure of F508del-CFTR function after low temperature rescue (27°C) of the mutant protein. After capturing baseline fluorescence reads, cells were acutely treated with L-arginine (1 mM) to activate SLC6A14 or vehicle, followed by CFTR activation with cAMP agonist forskolin (FSK 10 µM) or vehicle DMSO. Thereafter, CFTRinh-172 (10 µM) was added to all the wells. (d) Bar graph represents maximum change in ACC fluorescence from baseline ($\Delta F/F_0$) after acute addition of FSK, following low temperature (27°C) rescue of F508del-CFTR protein in split-open murine organoids (mean ± SEM). Paired t-test was performed (*p=0.045, ns = not significant, n = 4 mice for each genotype). (e) ACC assay performed on split-open colonic organoids from CF ($Cftr^{F508del/F508del}$) and double mutant ($Cftr^{F508del/F508del}$; $Slc6a14^{(-/y)}$) mice for CFTR channel function at physiological temperature (37°C). As above, SLC6A14 was activated with L-arginine (1 mM) or vehicle followed by CFTR stimulation by FSK (10 µM) or vehicle DMSO. All wells received CFTRinh-172 (10 µM) after activation to confirm the role for CFTR. (f) Bar graph represents maximum change in ACC fluorescence from baseline ($\Delta F/F_0$) after acute addition of FSK, at physiological temperature (37°C) in F508del-CFTR split-open murine organoids (mean ± SEM). Paired t-test was performed (****p<0.0001, ns = not significant, n = 3 mice for each genotype).

DOI: https://doi.org/10.7554/eLife.37963.012

The following figure supplements are available for figure 5:

**Figure supplement 1.** *Slc6a14* is a major arginine transporter in the colonic epithelium.

DOI: https://doi.org/10.7554/eLife.37963.013

**Figure supplement 2.** Split-open organoid model.

DOI: https://doi.org/10.7554/eLife.37963.014

**Figure supplement 3.** SLC6A14 does not enhance processing or cell surface stability of F508del-CFTR in BHK over-expression system.

DOI: https://doi.org/10.7554/eLife.37963.015

**Figure supplement 4.** SLC6A14 interaction with mutant F508del-CFTR.

DOI: https://doi.org/10.7554/eLife.37963.016

Altogether, these data show that *Slc6a14*-mediated arginine uptake is important for NO synthesis, and hence NO mediated signaling to the CFTR channel.

The effect of the nitric-oxide signaling pathway on PKA-mediated CFTR channel activation has not been studied extensively, but it has been shown that the down-stream signaling molecule, cyclic GMP (cGMP), does lead to enhanced PKG-dependent phosphorylation of CFTR (*Picciotto et al., 1992*; *Tien et al., 1994*). Hence, we were prompted to test the possibility that enhanced intracellular accumulation of NO will induce cGMP-dependent activation of CFTR. We employed the human colonic epithelial cell line (Caco-2) for these studies, as it is known to express functional CFTR (*Sood et al., 1992*). Using the ACC assay (*Ahmadi et al., 2017*), to measure CFTR channel function, we found that PKA-mediated activation of Wt-CFTR function can be enhanced by the NO donors GSNO and SNP, and by the downstream signaling molecule cGMP. This effect of GSNO on PKA-activated CFTR channel function is abolished in presence of protein kinase G (PKG) inhibitors Rp-8-pCPT-cGMPS and KT5823 (*Figure 7b,c,d*). Multiple inhibitors were employed to address possible concerns about inhibitor specificity (*Butt et al., 1994*; *Chiche et al., 1998*). Thus, we show that PKA-dependent channel activity of CFTR is enhanced by NO-mediated activation of PKG, in this colonic epithelial cell line, highlighting the importance of this signaling pathway for CFTR function in a model of human colon.

Next, to further study the impact of SLC6A14-mediated arginine transport on CFTR function, via the nitric oxide pathway (*Figure 7a*), we generated a Caco-2 cell line over-expressing *SLC6A14*. *SLC6A14* is not expressed in this cell line, so we over-expressed *SLC6A14-GFP* or *GFP* as a control (*Figure 7—figure supplement 1*). We previously confirmed that this carboxy terminally tagged protein retains function as an electrogenic, amino acid transporter (*Figure 7—figure supplement 2*). Using the CFTR functional assay described above, we found that CFTR function (measured as CFTRinh-172 response) was enhanced by pre-activation of SLC6A14 with arginine, only in the Caco-2 cell line over-expressing *SLC6A14*. This enhancement of CFTR function by SLC6A14-mediated arginine transport could be inhibited by PKG specific inhibitors Rp-8-pCPT-cGMPS and KT5823, as well as the iNOS inhibitor 1440W (*Figure 7e,f*). Taken together, we show that SLC6A14-mediated arginine transport can enhance CFTR function via the NO signaling pathway, in Caco-2 cells.

Finally, we tested the relevance of this signaling pathway in the mutant mice generated in our study. We asked if the reduced F508del-CFTR function measured in the double mutant mice could be overcome by bypassing the need for arginine uptake, and directly modifying F508del-CFTR protein by protein kinase G-dependent phosphorylation (*Figure 8*). The ACC assay was used to

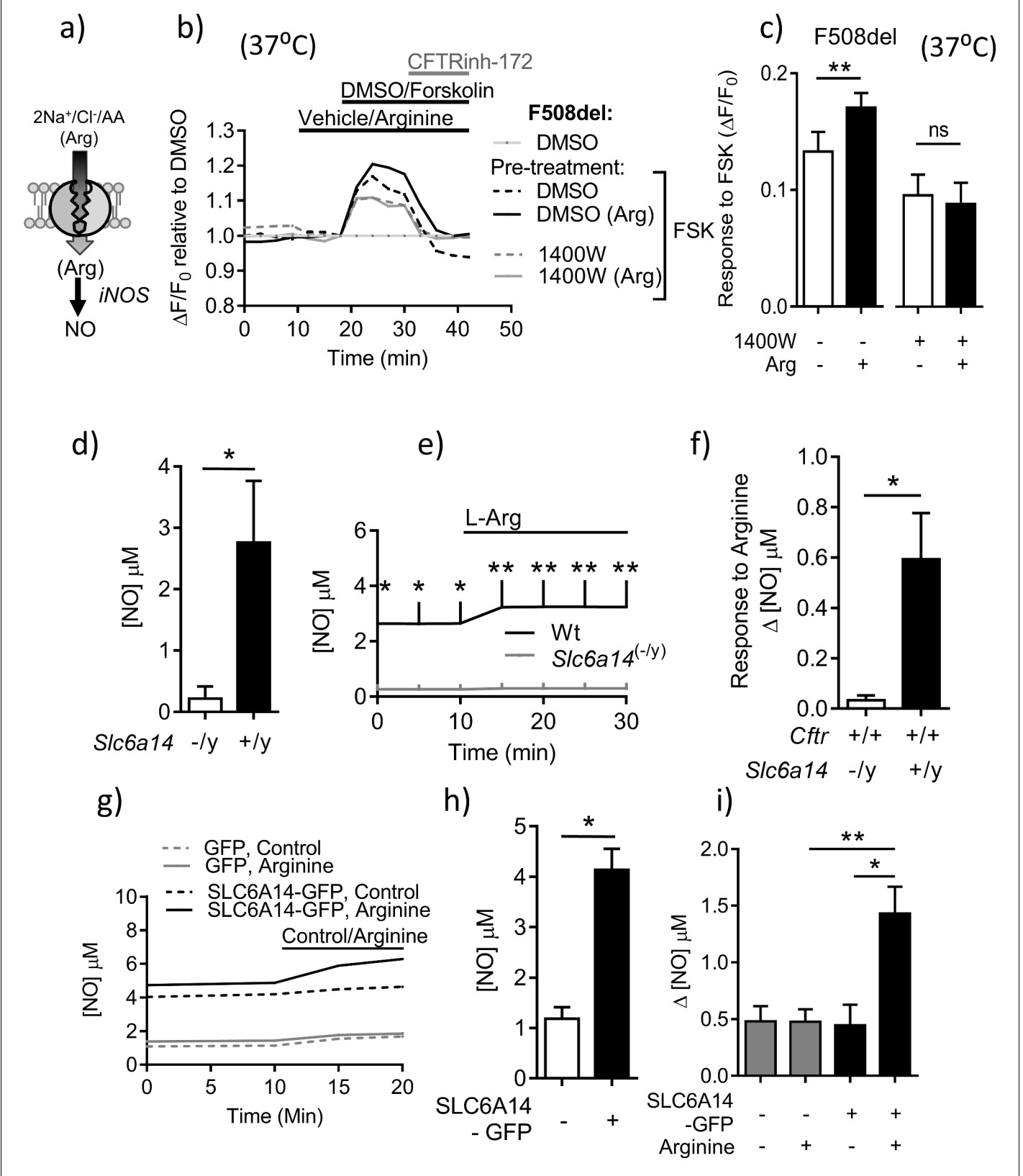

**Figure 6.** SLC6A14 expression enhances residual cAMP-dependent F508del channel function in murine intestinal tissues via arginine-mediated NO signaling. (a) Hypothetical model depicting the role of SLC6A14 in transporting arginine across the apical surface thereby increasing intracellular nitric

*Figure 6 continued on next page*

*Figure 6 continued*

oxide (NO) levels. (**b**) Split-open colonic organoids from CF (*Cftr*^F508del/F508del) mice were studied using the ACC assay at physiological temperature (37°C). Line graph represents change in fluorescence relative to baseline ($\Delta F/F_0$) as a measure of F508del-CFTR, after pre-incubation with vehicle or inducible Nitric-oxide Synthase (iNOS) inhibitor 1400W (100 µM) for 30 mins. (**c**) Bar graph represents maximum change in ACC fluorescence from baseline ($\Delta F/F_0$) after acute addition of FSK, at physiological temperature (37°C) in F508del-CFTR split-open murine organoids (mean ± SEM), after 30 mins pre-incubation with vehicle or iNOS inhibitor 1400W (100 µM). (\*\*p=0.006, ns = not significant, n = 4 biological replicates for each genotype). (**d**) Epithelial basal NO levels measured using DAF-FM fluorophore, by splitting open colonic organoids derived from Wt and *Slc6a14*^(-/y) mice. Bar graph represents mean ± SEM. Unpaired t-test was performed (\*p=0.022, n = 5 biological replicates). (**e**) Line graph represents change in NO levels upon acute addition of L-arginine (1 mM), in split-open colonic organoids derived from C57Bl/6 Wt and *Slc6a14*^(-/y) mice. Two-way ANOVA with Sidak's multiple comparison test was performed (p<0.0001, n ≥ 4 for each genotype, for t = 0, 5, 10 mins \*p<0.05, for t = 15, 20, 25 mins \*\*p=0.006) (**f**) Bar graph represents maximum change in intracellular NO levels upon acute addition of L-arginine (1 mM), in split-open colonic organoids from Wt and *Slc6a14*^(-/y) mice (mean ± SD). Unpaired t-test was performed (\*p=0.009, n ≥ 4 biological replicates for each genotype). (**g**) Line graph represents basal [NO] levels and change in [NO] after addition of SLC6A14 agonist L-arginine (1 mM) or control (buffer alone), in split-open murine FVB *Cftr*^F508del/F508del colonic organoids transduced with human SLC6A14-GFP or control GFP. (**h**) Bar graph represents basal [NO] levels in split-open murine organoids transduced with SLC6A14-GFP or just GFP. Mean ± SEM is plotted. Unpaired t-test was performed (\*p<0.0001, n = 3 biological replicates). (**i**) Bar graph represents change in [NO] levels (Δ[NO]) after addition of SLC6A14 agonist L-arginine (1 mM) or control (buffer alone), in split open FVB *Cftr*^F508del/F508del colonic organoids transduced with human SLC6A14-GFP or control GFP. Mean ± SEM is plotted. One-way ANOVA with Tukey's multiple comparison test was performed (\*p=0.02, \*\*p=0.006, n ≥ 3 biological replicates for each condition).

DOI: https://doi.org/10.7554/eLife.37963.017

The following figure supplements are available for figure 6:

**Figure supplement 1.** iNOS expression in primary murine colonic tissue.
DOI: https://doi.org/10.7554/eLife.37963.018
**Figure supplement 2.** NO-mediated signaling potentiates mutant F508del CFTR function in intestinal epithelial cells.
DOI: https://doi.org/10.7554/eLife.37963.019
**Figure supplement 3.** Standard curve for Nitric-Oxide (NO) measurement.
DOI: https://doi.org/10.7554/eLife.37963.020

measure F508del-CFTR function in colonic split-open organoids from the double mutant mice, after pre-treatment with 8Br-cGMP or vehicle (control). These studies were conducted at 27°C, given the very low levels of F508del-CFTR function that could be measured in colonic tissue derived from these double mutant mice. Interestingly, we found that pre-treatment of split-open organoids from double mutant mice, with membrane permeable cGMP (8Br-cGMP), enhanced forskolin-activated F508del-CFTR channel activity partially towards that observed in tissues from F508del mice with native SLC6A14 expression (*Figure 8a,b*). As the organoid swelling assay is known to be very sensitive in detecting F508del-CFTR-mediated fluid secretion (*Dekkers et al., 2016*; *Dekkers et al., 2013*), we were encouraged to assess the effect of 8Br-cGMP on forskolin-mediated fluid secretion in colonic organoids from the double mutant mice, at 37°C. Importantly, the partial rescue effect induced by 8Br-cGMP pre-treatment was recapitulated at physiological temperature in the organoid swelling assay (*Figure 8c,d*). Specifically, pre-treatment of organoids from double mutant mice with cGMP restored forskolin-induced swelling (FIS) to approximately 50% of the swelling measured in F508del-CFTR organoids, at 37°C. Together, these findings support the hypothesis that the arginine transport activity of SLC6A14 and the downstream signaling via NO regulated cGMP plays an important role in modifying F508del-CFTR channel function and F508del-CFTR-mediated secretion.

## Discussion

This work provides a mechanism whereby *SLC6A14*, previously identified as a genetic modifier of CF disease severity, can ameliorate the primary CF-causing defect. We found that SLC6A14 is a major arginine transporter on the apical surface of the colonic epithelium and in mice modeling the major CF-causing mutation F508del, and its arginine transport function can ameliorate the basic defect in fluid secretion.

*SLC6A14* has been identified as a genetic modifier of multiple CF disease outcomes, including meconium ileus (*Sun et al., 2012*), lung disease (*Corvol et al., 2015*) and the age of first infection by *Pseudomonas aeruginosa* (*Li et al., 2014*). This pleiotropic effect supports the hypothesis that it modifies the basic defect responsible for disease in multiple organs, namely the defect in anion conduction. The work in this study focused on the impact of disrupting *Slc6a14* on the CF phenotype in

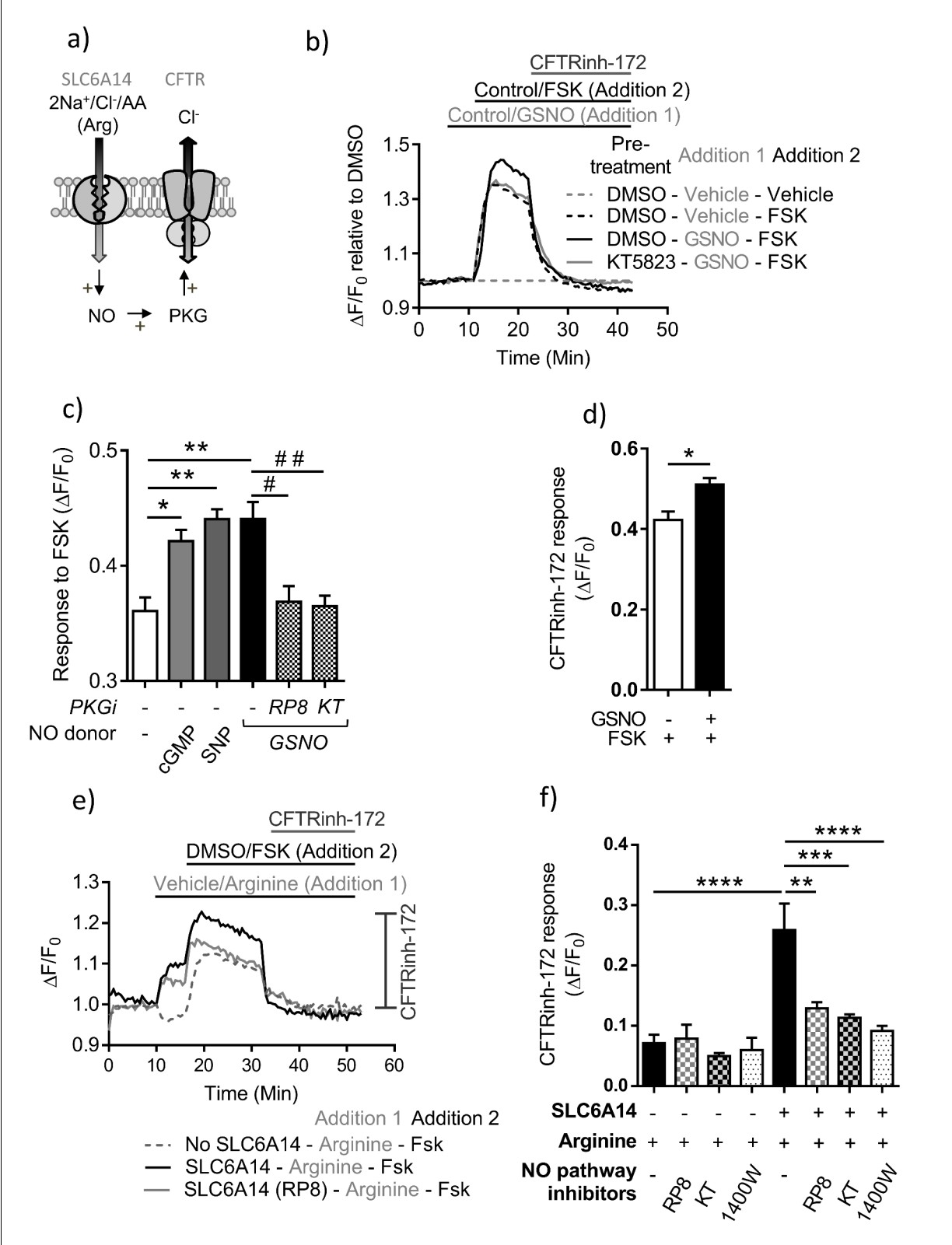

**Figure 7.** PKA-dependent channel activity of CFTR is enhanced by NO dependent activation of PKG in colonic epithelium. (a) Model depicting the hypothesis that SLC6A14-mediated arginine uptake across the epithelium would lead to an increase in intracellular NO levels, which would then potentiate CFTR channel function via PKG activation. (b) Line graph represents CFTR function measured using the FLIPR® membrane potential assay in Caco-2 colonic intestinal epithelial cells. Cells were pre-treated with PKG inhibitor KT5823 (1 μM) or DMSO control. After reading baseline intracellular

*Figure 7 continued on next page*

Figure 7 continued

NO levels were increased using NO donor GSNO (10 μM) or vehicle control, followed by cAMP activation of CFTR using FSK (10 μM) or DMSO vehicle. (c) Bar graph represents response to FSK from baseline ($\Delta F/F_0$), in Caco-2 cells pre-treated with PKG inhibitors Rp-8-pCPT-cGMPS (RP8), KT5823 or vehicle DMSO, followed by addition of NO donor GSNO as shown in *Figure 7b*. The effect on FSK response by another NO donor SNP (50 μM) and direct increase of intracellular cGMP (10 μM) was also studied in this system. Mean ± SEM are plotted. One-way ANOVA with Tukey's multiple comparison test was performed (*p=0.006, **p=0.0001, # p=0.004, # # p=0.002, n = 3 biological replicates). (d) Bar graph represents CFTRinh-172 (10 μM) response, elicited at the end of the experiment after full CFTR activation. Mean ± SEM are plotted. Unpaired t-test was performed (*p=0.007, n = 3 biological replicates). (e) Line graph represents CFTR function measured using the FLIPR® membrane potential assay in Caco-2 epithelial cells transduced with *SLC6A14-GFP* or control *GFP* alone. Cells were pre-treated with PKG inhibitors Rp-8-pCPT-cGMPS (10 μM), KT5823 (1 μM), or iNOS inhibitor 1400W (50 μM) or DMSO control. After reading for baseline, SLC6A14 was stimulated using L-arginine (1 mM) or vehicle control, followed by cAMP activation of CFTR using FSK (10 μM) or DMSO vehicle. This was followed by addition of CFTRinh-172 (10 μM). (f) Bar graph represents CFTRinh-172 (10 μM) response, elicited at the end of the experiment after full CFTR activation, in Caco-2 cells over-expressing *SLC6A14-GFP* or control *GFP*. Mean ±SEM are plotted. One-way ANOVA with Tukey's multiple comparison test was performed (****p<0.0001, **p=0.003, ***p=0.0001, n = 3 biological replicates).

DOI: https://doi.org/10.7554/eLife.37963.021

The following figure supplements are available for figure 7:

**Figure supplement 1.** *SLC6A14-GFP* over-expression in Caco2 colonic epithelial cells.

DOI: https://doi.org/10.7554/eLife.37963.022

**Figure supplement 2.** SLC6A14-GFP functions as an electrogenic amino-acid (arginine) transporter.

DOI: https://doi.org/10.7554/eLife.37963.023

the mouse intestine, because this phenotype has been well described by multiple laboratories (*Canale-Zambrano et al., 2007*; *Rozmahel et al., 1996*; *Scholte et al., 2004*; *van Doorninck et al., 1995*; *Wilke et al., 2011*). In support of the above hypothesis, we found that the intestinal phenotypes associated with the primary CF defect in mice, namely defective secretory capacity with thickened smooth muscle, were worsened by disruption of *Slc6a14* expression.

Interestingly, disruption of *Slc6a14* expression in mice bearing Wt-CFTR did not cause profound changes in colonic morphology (*IMPC, 2018*), function or mouse survival. These findings confirm previous findings reported by Ganapathy and colleagues, who disrupted *Slc6a14* and did not observe gross changes in mouse health (*Babu et al., 2015*). However, disruption of *Slc6a14* did impair arginine uptake across the apical membrane of colonic epithelium in Wt-*Cftr* C57Bl/6 mice (*Figure 1e*), and impaired cAMP-mediated fluid secretion (*Figure 4—figure supplement 1*). Together, these observations support the hypothesis that this arginine transporter on the apical membrane of colonic epithelium has a dominant role in CFTR regulation rather than nutrient uptake, as amino acid uptake normally occurs in the small intestine.

Most of our studies of the CF phenotype employed transgenic mice engineered to express the F508del-*Cftr* mutation. In contrast to the consequences of the F508del mutation in the human CFTR protein, the negative consequences of this mutation in folding and assembly of the mouse CFTR protein are somewhat less severe, hence there is residual epithelial expression and function being detected in this organism (*Scholte et al., 2004*; *van Doorninck et al., 1995*; *Wilke et al., 2011*). As previously published, this reflects a less deleterious effect of the mutation on the folding, assembly and maturation, such that a significant amount of the mutant mouse protein (still less that wild-type) will reach the surface and function (*French et al., 1996*; *Rozmahel et al., 1996*; *Scholte et al., 2004*; *van Doorninck et al., 1995*). It was this residual F508del-*Cftr* function that was modulated by *Slc6a14* in the current study. We predict that the extent of modulation by SLC6A14 in the colonic epithelium of CF patients with the F508del mutation will depend on its residual expression levels at the apical membrane of this epithelium, or the abundance of this mutant at the cell surface after correction with Lumacaftor.

Our findings suggest that the modulation of F508del-CFTR by SLC6A14 is dependent, at least in part, on arginine-mediated NO production and PKG activation. The role of the arginine-NO pathway has been well studied in models of CF (*Grasemann et al., 2011*; *Grasemann et al., 1997*; *Grasemann and Ratjen, 2012*; *Grasemann et al., 2005*; *Oliynyk et al., 2013*; *Poschet et al., 2007*; *Scott et al., 2015*). At the molecular level, it is known that the downstream regulators of this pathway, cGMP and PKG, can enhance wild-type and mutant CFTR channel function and trafficking (*Golin-Bisello et al., 2005*; *Leier et al., 2012*; *Poschet et al., 2007*; *Vaandrager et al., 1997*).

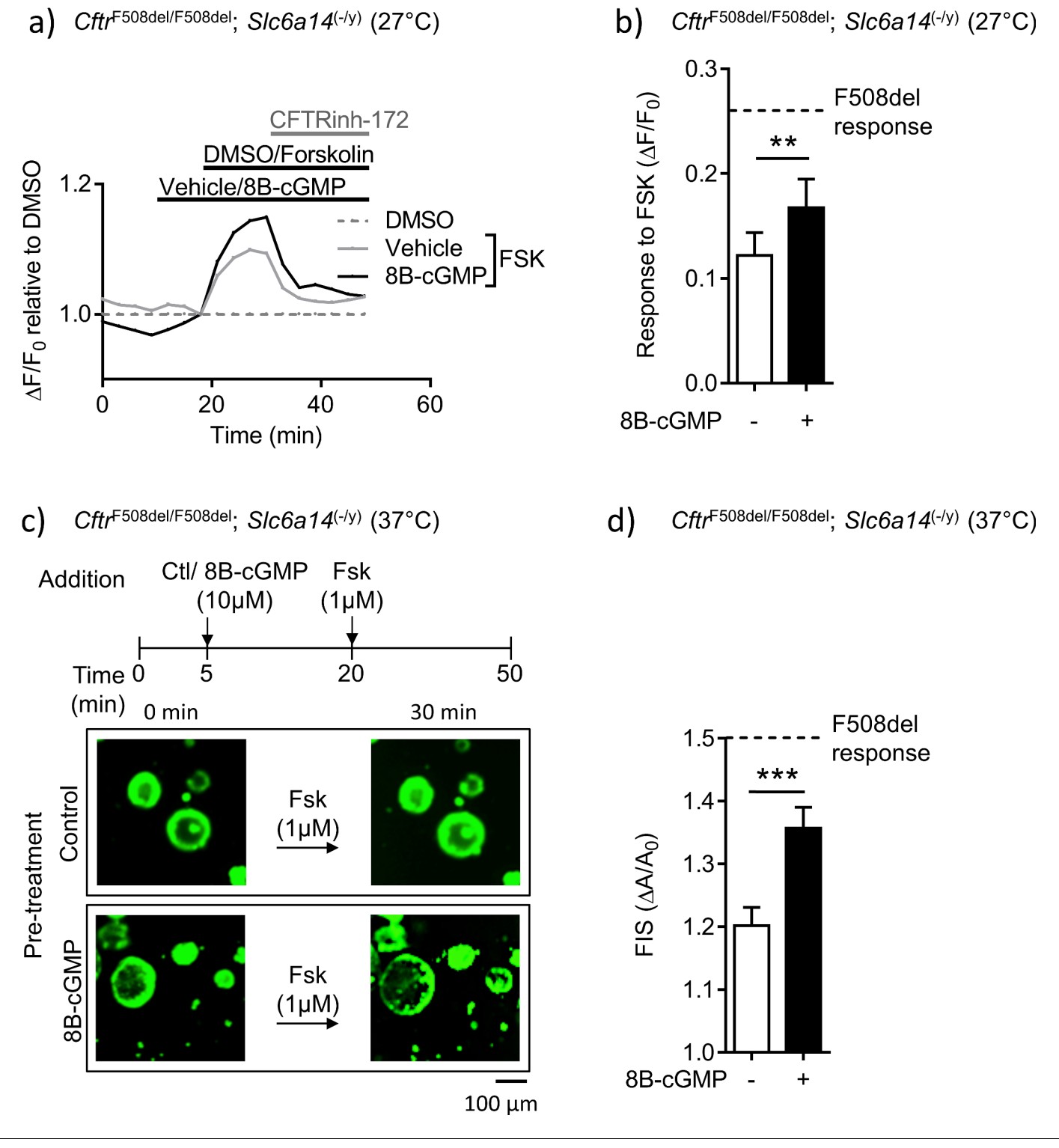

**Figure 8.** Mutant F508del-CFTR function in double mutant murine organoids can be enhanced by directly enhancing PKG-mediated phosphorylation. (a) Split-open organoids from double mutant (*Cftr*$^{F508del/F508del}$; *Slc6a14*$^{(-/y)}$) mice were used to perform ACC assay, after low temperature (27°C) rescue of the mutant F508del CFTR protein. Line graph represents change in fluorescence from baseline ($\Delta F/F_0$) relative to DMSO vehicle addition. After capturing baseline read, cells were acutely treated with 8Br-cGMP (10 µM) or vehicle, followed by addition of CFTR cAMP agonist FSK (10 µM). All wells received CFTRinh-172 (10 µM) at the end. (b) Bar graph represents maximum change in ACC fluorescence from baseline ($\Delta F/F_0$) after acute addition of FSK, following low temperature (27°C) rescue of F508del-CFTR protein in double mutant spilt-open organoids (mean ± SEM). Stippled line represents

*Figure 8 continued on next page*

*Figure 8 continued*

ACC response to FSK in temperature rescued (27°C) CF ($Cftr^{F508del/F508del}$) split-open murine organoids, in absence of 8Br-cGMP. Paired t-test was performed (**p=0.0029, n = 5 mice for each genotype). (c) Organoids from double mutant ($Cftr^{F508del/F508del}$; $Slc6a14^{(-/y)}$) mice were stained with Calcein AM and after capturing baseline images, organoids were acutely treated with 8Bromo-cGMP (8Br-cGMP 10 µM) for 15 mins, followed by FSK (1 µM) for 30 mins. Representative images of forskolin-induced swelling (FIS) are shown for organoids pre-treated with 8Br-cGMP or vehicle (control). Experiments were performed at physiological temperature of 37°C. (d) Bar graph represents FIS response in double mutant ($Cftr^{F508del/F508del}$; $Slc6a14^{(-/y)}$) murine organoids, as measured by change in area from baseline after 30 mins of FSK addition ($\Delta A/A_0$). Stippled line represents FIS response in CF ($Cftr^{F508del/F508del}$) organoids in absence of 8Br-cGMP. Graph represents mean ± SEM. Experiments were performed at physiological temperature of 37°C. Unpaired t-test was performed (***p=0.0005, n = 4 mice).

DOI: https://doi.org/10.7554/eLife.37963.024

Previous studies of mouse derived intestinal organoids showed that agonists of PKG are effective in enhancing CFTR-mediated organoid swelling, in the absence of agonists of the canonical regulation of CFTR, such as PKA (*Pattison et al., 2016*; *Picciotto et al., 1992*; *Seidler et al., 1997*; *Vaandrager et al., 1997*). Our studies confirm the functional significance of this regulatory pathway in murine colon, and further, our findings suggest that PKG-mediated phosphorylation exerts a 'priming' effect on the canonical PKA-mediated activation of CFTR. There are five sites at which CFTR can be phosphorylated by PKG (NetPhosK v3, consensus strength >0.5). Two are localized in the regulatory 'R' domain, at position 788 and 795, which overlap with the consensus site for PKA phosphorylation (*Pasyk et al., 2015*). Interestingly, the strength of the consensus sites for PKG at position 788 is predicted to exceed that of PKA, and may be preferentially phosphorylated by PKG.

Both of our assays of regulated F508del-CFTR channel function in colonic tissue lacking SLC6A14 showed that cGMP partially rescued the defects in cAMP-induced channel activation and organoid swelling. However, while significant, this augmentation did not fully restore these functions to those observed in tissues obtained from F508del mice (expressing native *Slc6a14*). Hence, there could be other, yet unidentified, signaling pathways involved in SLC6A14-mediated CFTR modulation. Alternatively, the doubly mutated mice may harbor secondary, chronic consequences, which impact F508del-*Cftr*, and we will assess such consequences in future studies.

The current findings suggest that upregulation of *Slc6a14* expression and SLC6A14-mediated arginine uptake will ameliorate the disease phenotype associated with *Cftr* mutation. Interestingly, studies by Grasemann and colleagues showed that patients with CF have a decreased concentration of exhaled NO (*Grasemann et al., 1997*), related to decreased nitric-oxide synthase (iNOS) activity, increased cellular arginase activity, and decreased arginine bioavailability (*Grasemann et al., 2011*; *Grasemann and Ratjen, 2012*; *Grasemann et al., 2005*). The mechanism underlying decreased arginine bioavailability in CF is unknown, but we propose that this deficit worsens the channel function of the low, residual levels of F508del-CFTR and that increased SLC6A14 expression will offset this deficit.

An increase in SLC6A14-mediated arginine uptake would be expected to overcome decreased arginine bioavailability in CF tissues, and enhance NO and cGMP-mediated 'priming' of F508del-CFTR channel activity at the cell surface. Although not specifically tested in this study, we showed in primary cultures of airway epithelium derived from CF patients that bacterial products enhance the expression of *SLC6A14* (*Di Paola et al., 2017*). As the bacterial load in the colon of patients will be high (*Pereira and Berry, 2017*), we expect that SLC6A14 expression will be upregulated and poised to modify any mutant CFTR protein that exhibits residual channel function. Finally, it has yet to be determined if this modification by SLC6A14 is dependent on the genotype of GWAS-identified SNPs.

## Materials and methods

### *Slc6a14* mouse genotyping

DNA was extracted from clipped tails after overnight digestion with Proteinase K. Polymerase chain reaction (PCR) was performed using the KAPA HotStart mouse genotyping kit, for 35 cycles. 2 sets of primers were used for amplification; the first set amplified the region in intron 5 of *Slc6a14* gene,

**Table 1.** Genotyping primers

| Primer set | Forward (5'–3') | Reverse (5'–3') |
|---|---|---|
| 1 | TTCAAGTCTCTCTAGCTTCAGGTC | TTATCTGGTAGCTTCCTGTGACTC |
| 2 | CCATTACCAGTTGGTCTGGTGTC | AAGGTGCTTATTTGAACTGATGGCGAGC |

DOI: https://doi.org/10.7554/eLife.37963.025

which is present in wild-type mice; second set of primers amplified the lacZ cassette present in the *Slc6a14* knock-out mice. The primer sequences of the two sets are in *Table 1*.

## Quantitative real-time PCR

Murine organs (as listed in *Figure 1*) were collected and immediately placed in RNAlater® (Ambion) at room temperature. Samples were later stored at 4°C until further use. Within 1 week, samples were homogenized using Rotor-TissueRuptor (Qiagen) and mRNA was extracted using RNeasy® Plus Mini Kit (Qiagen), following listed instructions. After determining the purity and yield of the RNA spectrophotometrically, all samples were immediately stored at −80°C. Samples were used with a 260/280 ratio of 2.0–2.2 and concentrations higher than >100 ng/μL. 1 μg of total mRNA from each sample was reverse transcribed into cDNA using iScript™ cDNA Synthesis Kit (BioRad) per the manufacturer's instructions. For real-time PCR (qRT-PCR), expression levels were measured using primers listed in *Table 2*, on the CFX96 Touch™ Real-Time PCR Detection System (BioRad) using the Eva-Green fluorophore (SsoFast EvaGreen Supermix with Low Rox, BioRad).

A similar protocol provided by the manufacturer (Qiagen) was used for mRNA expression analysis in Caco-2 colonic epithelium cell line. Cells were lysed using the lysis buffer provided in the kit, and expression analysis was performed as described above using primers listed in *Table 3*.

## Mouse colonic organoid culture

As described previously (*Dekkers et al., 2013*; *Sato et al., 2009*), murine colonic tissue was obtained from mice on FVB or C57BL/6N background, aged 7–8 weeks old. 3–4 cm of colonic tissue was cut into pieces approximately 0. 5 cm$^2$ in size, and washed with phosphate buffered saline (PBS, Wisent). Colonic crypts were then incubated in PBS containing 10 mM EDTA for 30 mins at 4°C. Colonic crypts were removed from the epithelium through mechanical stress and pelleted through centrifugation. Pelleted crypts were then washed once with ice cold PBS to remove excess debris from the primary colonic tissue. Pelleted crypts were re-suspended in 100% Matrigel (Corning) and seeded at approximately 10 crypts/μL of Matrigel per well. The Matrigel dome was allowed to dry for 30 min at 37$^0$C. Once solidified, growth factor conditioned medium was added. Growth factor conditioned medium was changed every second day for optimal organoid formation.

Organoids were pre-incubated with a fluorescent live-cell marker, Calcein-AM (20 μM) for 30 mins and were removed from the Matrigel matrix by washing the Matrigel dome with ice cold Hank's Balanced Salt Solution (HBSS, Wisent). Organoids were plated onto a 96-well plate and were centrifuged at 500 *g* at 4$^0$C to settle. Organoids were imaged for 5 mins to establish baseline. CFTR was stimulated with FSK (1 μM) . CFTR-mediated fluid secretion measured as FSK-Induced Swelling (FIS) was captured by imaging for 30 mins at 5 min intervals, following the addition of FSK, with bright field and fluorescence microscopy, as previously described (*Dekkers et al., 2013*). Analysis was performed using Cell Profiler v2.1 (*Figure 4—figure supplement 2*) and ImageJ v1.6.

**Table 2.** Primers used for qRT-PCR (mouse)

| Gene | Forward (5'–3') | Reverse (5'–3') |
|---|---|---|
| *Cftr* | CGGAGTGATAACACAGAAAGT | CAGGAAACTGCTCTATTACAGAC |
| *Tbp* | CAAACCCAGAATTGTTCTCCTT | ATGTGGTCTTCCTGAATCCCT |
| *Gusb* | CCGATTATCCAGAGCGAGTATG | CTCAGCGGTGACTGGTTCG |
| *Slc6a14* | GCTTGCTGGTTTGTCATCACTCC | TACACCAGCCAAGAGCAACTCC |

DOI: https://doi.org/10.7554/eLife.37963.026

**Table 3.** Primers used for qRT-PCR (human)

| Gene | Forward (5'−3') | Reverse (5'−3') |
|------|-----------------|-----------------|
| SLC6A14 | TATGGCGCAATTCCATACCC | CCAGGTATGGACCCCAGTTA |
| GUSB | CCCATTATTCAGAGCGAGTATG | CTCGTCGGTGACTGTTCAG |

DOI: https://doi.org/10.7554/eLife.37963.027

## Organoid transduction

Murine colonic organoids were cultured for 5–7 days. Organoids were removed from the supporting Matrigel with ice cold PBS washes and pelleted through centrifugation at 500 *g* for 5 min at 4°C. Organoids were re-embedded in Matrigel, while being transduced with lentivirus vector packaging *LV-SLC6A14-GFP* or *LV-GFP* at MOI 50 (Multiplicity of infection 50). The GFP tag was inserted on the carboxy terminus of SLC6A14 and we confirmed that its functional expression as an electrogenic arginine transporter is not disrupted by this modification in *Figure 7—figure supplement 2*. Medium containing lentivirus particles was removed after 24 hr. Media was changed every day for 2 days. Organoids were stained for GFP and E-cadherin, to confirm transduction and *SLC6A14-GFP* expression using immunofluorescence studies as described below. Imaging was performed with Olympus Quorum Spinning Disk Confocal Microscope (SickKids Imaging Facility).

## Immunofluorescence

Samples were fixed with methanol for 10 mins. Cells were washed three times with PBS, 5 mins per wash. After washes, sample were then washed and blocked in 4% Bovine Serum Albumin (BSA) in PBS for 30 mins and incubated with the appropriate primary antibody combinations against GFP (Abcam), E-cadherin (Cell Signaling), and DAPI (ThermoFisher Scientific) or antibody against apical membrane marker Zona Occluden-1 (ZO-1, Thermo Fisher Scientific) and DAPI, overnight at room temperature. Primary antibody was washed away with PBS and samples were washed three times with PBS, 5 mins per wash. Samples were then incubated with monoclonal or polyclonal secondary antibodies (Life Technologies) for 1 hr. Samples were imaged using Nikon A1R Confocal Microscope and NIS elements and Volocity 6.3 software.

## Mass spectrometry

Whole tissue extracts were lysed using a buffer containing 50 mM Tris-HCl, pH 7.4, 150 mM NaCl, 1 mM EDTA, containing 0.1% (vol/vol) SDS, 0.1% (vol/vol) Triton X-100, 2% (vol/vol) protease inhibitor mixture (Amresco) and 1x phosphatase inhibitor cocktail (Roche), and then incubated at 4°C for 15 mins. Samples were centrifuged for 15 mins at 18,000 g and supernatants were collected for analysis, as previously described (*Grasemann et al., 2011*). From these samples, amino acids were quantified by LC/MS/MS at the Analytical Facility for Bioactive Molecules (The Hospital for Sick Children, Toronto, Canada). Tissue lysates (50 µL), standards, QC's, and deuterated internal standards were mixed with 1 mL of methanol and then centrifuged at 20,000 *g*. Supernatants were transferred to a conical tube and taken to dryness under a gentle stream of nitrogen. Samples were then derivatized with butanol in 3N HCl for 15 mins at 65°C, then dried again and reconstituted in 1:9 ratio of water to acetonitrile 5 mM ammonium formate (pH 3.2), and analyzed by LC/MS/MS. Samples were injected onto a Kinetex 2.6 u HILIC 50 × 3.0 mm column (Phenomenex) on an Agilent 1290 LC system coupled to a Sciex Q-Trap 5500 mass spectrometer. Samples were eluted using a gradient of A) 9:1 water to acetonitrile 5 mM ammonium formate (pH 3.2) and B) 1:9 water to acetonitrile 5 mM ammonium formate (pH 3.2) over six mins. Data were collected and analysed using Sciex Analyst v1.6.3.

## IGF-1 level measurements

Serum samples were obtained from adult mice and kept at room temperature for 2 hr. The sera were collected following centrifugation (2000 *g* for 20 mins) and frozen immediately at −80°C until analysis. The Quantikine ELISA Mouse/Rat IGF-1 Kit (MG100, R and D Systems, USA) was used, which utilizes a double-antibody sandwich enzyme-linked immunosorbent assay (ELISA), to determine the level of mouse insulin growth factor-1 (IGF-1). Assay procedure was followed as specified

by the manufacturer. The absorbance was measured at 450 nm using the i3x-Spectrophotometer (Molecular Devices, USA). Data were analyzed using GraphPad Prism v6.01.

## Serum amino acid analysis

Serum samples collected from adult mice at day 56, as described above. As previously described (*Di Paola et al., 2017*), amino acid content was determined via HPLC, with a reverse-phase C18 column (Courtesy SPARC, The Hospital for Sick Children, Toronto).

## Histology

Samples were fixed in formalin before being submitted for standard histological processing. Paraffin-embedded sections were stained with Alcian blue for mucus, at the Mount Sinai Toronto Center for Phenogenomics (TCP) Pathology Department. These slides were scanned (SickKids Imaging Facility) and sections were analyzed using 3DHISTECH software. Annotations were drawn along the perceived borders of tissue layers (*Figure 1* and *Figure 3—figure supplement 1*). This was repeated for a total of 3 sections per slide per mouse. All the algorithms used by the software had fixed parameters for all replicates and conditions. Data were exported in excel file and statistics was performed using GraphPad Prism v.6.01.

## Morphometric measurements

Mice weights were measured (American Weigh Scales Black Blade Digital Pocket Scale) and tracked 2–3 times a week for 8 weeks, starting from day 9. Length of the body (tip of nose to base of tail-crown rump length) was also tracked for all mice.

## In vivo fluid secretory assay

All animal studies were performed after ethics approval of Animal Use Protocol. C57BL/6N and FVB mice strains were used for the experiment. Protocol was adapted and modified from a previously described method (*Sonawane et al., 2006*). Mice were fasted 15 hr before the experiment and given colyte + 5% dextrose. Next day, they were anesthetized using 2–3% of isoflurane for maintenance; up to 4–5% for induction, with oxygen from a precision vaporizer. Mice were given buprenorphine (0.01 mg/kg) as an analgesic before surgery. A heating pad was used to maintain body temperature. Mice were placed on dorsal recumbent position throughout the procedure. The ventral abdominal wall was shaved using a surgical electric shaver. Antiseptic scrubbing of the abdominal wall was done with 70% ethanol followed by a solution of 10% povidone-iodine. After sterile draping, a ventral incision along the left midclavicular line was made through the skin, subcutaneous tissue and anterior abdominal muscle layers. This was followed by an incision through the peritoneum, to gain access to the peritoneal cavity. After visual inspection, caecum was identified and a ligature was placed, using a perma-hand silk (5-0; black braided; Ethicon) near the caeco-colic junction. Another ligature was made about 1 cm from the first ligature, to form a closed loop of 1 cm in length. A third and fourth ligature was made 1 cm from the second ligation, to form two closed loops side by side. Care was taken to prevent damage to mesenteric vessels and ischemia of the intestinal loop. Loops were injected with test drugs dissolved in 50 µL of HBSS (without glucose). The test drugs included control (DMSO), FSK (10 µM) and IBMX (100 µM). Mice were properly sutured with coated Vicryl suture (5-0; undyed braided; Ethicon) and let recover in a clean housing cage under a warm pad. Mice were monitored closely during recovery period, for any signs of distress. Mice were checked every 30 min during the 1 hr time, after which they were euthanized using $CO_2$ and the loops were removed at the end of the hour. Loop length and weights were measured to quantify fluid secretion.

## Ex vivo closed loop amino acid uptake assay

Mice were housed in a pathogen-free environment in the SickKids Laboratory Animal Services (LAS) facility. All experiments were performed under the SickKids Animal Care and Use Committee approved protocols. As previously published using the tracheal loops, we applied a similar methodology on intestinal loops (*Di Paola et al., 2017*). Male mice between the ages of 6-8 weeks were sacrificed. Colon or ileum of mice were isolated, and three closed loops each measuring 1.5 cm were formed using silk sutures (Ethicon). Each loop was injected with 100 µL of buffer (25 mM HEPES, 140

mM sodium chloride, 5.4 mM potassium chloride, 1.8 mM calcium chloride, 0.8 mM magnesium sulfate and 5 mM glucose; pH 7.4; 300 mOsm) supplemented with 1 µCi/mL of L-[2,3-$^3$H]-arginine (specific activity of 54.6 Ci/mmol), and 100 µM or 20 mM cold L-arginine. The concentrations of sodium and chloride in this buffer exceed the Km's required for transporter activity and likely result in maximum transport via SLC6A14 (*Anderson et al., 2008*; *Karunakaran et al., 2011*).

After waiting 15 mins, the loops were opened, and the apical surfaces were flushed with ice-cold buffer (as above) supplemented with 20 mM arginine and lysed with 0.5M NaOH. Following lysis on ice for approximately 1 hr, samples were spun down at 18,000 *g* for 10 mins. 12.5 µL of each lysate was added to 2 mL of EcoScint A Scintillation Fluid (Diamed, Switzerland) along with the appropriate controls, and counts were read using a Beckman Scintillation Counter (LS-6000IC). Total lysate protein was determined using a BioRad protein assay. Amount of protein was determined by absorbance (at 595 nm) of a Coomassie Brilliant Blue G-250 dye. 10 µL of diluted protein samples (diluted 1:10 and 1:20 in ddH$_2$O) was added to 200 µL of dye, and protein concentration was interpolated using a bovine serum albumin (BSA) standard curve. Normalized arginine uptake was calculated by dividing scintillation counts per minute (CPM) by the determined protein concentration. The final calculation units used were in CPM per mg/mL of protein.

## Studies of CFTR and F508del-CFTR-mediated swelling of colonic organoids

Organoids were pre-incubated with a fluorescent live-cell marker, Calcein-AM (20 µM) for 30 min and were removed from the Matrigel matrix by washing the Matrigel dome with ice cold buffer. Organoids were plated into 96-well plate and were centrifuged at 500 *g* and 4°C to settle. Organoids were imaged for five mins to establish baseline. CFTR was stimulated with FSK (1 µM). CFTR-mediated fluid secretion measured as FSK-Induced Swelling (FIS) was captured by imaging for 30 mins at 5 min intervals following the addition of FSK, with bright field and fluorescence microscopy (Nikon Epifluorescence/Histology Microscope), as previously described (*Dekkers et al., 2013*). Analysis was performed using Cell Profiler v2.1 (*Figure 4—figure supplement 2*). Fluorescent images extracted from FIS videos were exported as .TIFF (Tagged Image File Format) files and a pipeline using CellProfiler (Carpenter Lab) was created to analyze the video files. Following manual thresholding (0.1) of background fluorescence and identifying and tracking primary objects (>50 organoids/well) within the diameter limit of 40–150 pixels, the CellProfiler pipeline generated each identified objects' area, perimeter, diameter, and radius changes. The change in area ($\Delta A$) relative to baseline ($A_0$) was calculated, and the maximum change in $\Delta A/A_0$ within 30 mins of FSK addition was used as a measure of FIS (*Dekkers et al., 2013*).

## Generation of split open organoids

The organoids were suspended in 10 mL of DMEM/F12 (Dulbecco's Modified Eagle Medium with nutrient mixture F12) media and spun down at 500 *g* for 5 mins. The organoids pellet was then resuspended in conditioned media (*Dekkers et al., 2013*) and plated on 96-well clear bottom plates, coated with Poly-L-lysine (0.01% solution), by pipetting 40 µL of solution in to each well for 5 mins at room temperature, while shaking the plate on an orbital shaker, then removing the excess fluid after 5 mins and washing the wells with PBS. The wells were then allowed to dry in a sterile environment for 2 hr. Thereafter, 200 µL/well of the resuspended organoid culture was pipetted into each well. After 24 hr (Day 2), 100 µL of media was removed from each well and replaced with 100 µL of fresh conditioned media. This promotes spontaneous splitting of the organoids to form a lawn of cells such that the apical membrane is exposed to the medium as shown in *Figure 5—figure supplement 2*. On Day 3, fluorescence-based assays for nitric-oxide production or CFTR channel function were performed using DAF-FM or ACC assay respectively.

## Apical Chloride Conductance (ACC) Assay for CFTR in 'split-open' organoids

Murine organoids were generated as previously described (*Dekkers et al., 2013*; *Sato et al., 2009*). For splitting-open the organoids, they were suspended in 10 mL of DMEM/F12 medium and spun down at 500 *g* for 5 mins. The organoids pellet was then resuspended in conditioned medium (*Dekkers et al., 2013*), and then plated on 96-well clear bottom plates. The wells of this plate were

coated with Poly-L-lysine (0.01% solution, as described above). Thereafter, 200 µL/well of the resuspended organoid culture was pipetted into each well. After 24 hr (Day 2), 100 µL of media was removed from each well and replaced with 100 µL of fresh conditioned media. On Day 3, the ACC assay was performed. After 48 hr, split-open 2D lawns were produced, giving access to apical membrane. FLIPR® based ACC assay was then performed on the apically exposed colonic epithelium (*Ahmadi et al., 2017*; *Molinski et al., 2015*). Briefly, 0.5 mg/ml of the blue membrane potential dye was dissolved in buffer (112.5 mM NMDG- Gluconate, 36.25 mM NaCl, 2.25 mM K.Gluconate, 0.75 mM KCl, 0.75 mM $CaCl_2$, 0.5 mM $MgCl_2$ and 10 mM HEPES, 300 mOsm, pH 7.35). The low NaCl concentration was chosen to impose an outward electrochemical driving force for chloride while maintaining sodium and chloride concentrations above their Km values for SLC6A14 activity (*Karunakaran et al., 2011*). After 30 min incubation at 27 or 37°C, the plate was transferred to a micro-plate reader (Molecular Devices Paradigm). Fluorescence was read using excitation wavelength of 530 nm and emission wavelength of 560 nm. Multiple points in each well were read, and after capturing at least three baseline reads, pharmacological modulators were added (2.5 µL/well). The concentrations of sodium and chloride in this buffer exceed the Km's required for transporter activity (*Karunakaran et al., 2011*) and likely result in maximum transport via SLC6A14 (*Karunakaran et al., 2011*). The chloride concentration used in the assay is within the range of what would be present in the colon (*Schilli et al., 1982*). Data were analyzed as previously described (*Ahmadi et al., 2017*), and statistics were performed using GraphPad Prism v6.01.

## Nitric oxide measurements in 'split-open' organoids

Colonic organoids from mice were grown in a three-dimensional matrix and then acutely (72 hr) split-open in 96-well plates as described above. The organoids were incubated in DMEM-Dulbecco's Modified Eagle Medium (ThermoFisher Scientific, USA) with 1 mM L-arginine (Sigma-Aldrich, USA). To prepare the organoids for nitric oxide measurements, 200 µL of NaCl buffer (145 mM NaCl, 3 mM KCl, 3 mM $CaCl_2$, 2 mM $MgCl_2$, 10 mM HEPES) containing 7 µM DAF-FM Diacetate (4-Amino-5-Methylamino-2',7'-Difluorofluorescein Diacetate, Cayman Chemical, USA) and 10 µM Calcein Blue AM (ThermoFisher Scientific, USA) was added to each well for 1 hr. The organoids were then washed three times with PBS, to wash the dye. The organoids were then kept in the aforementioned NaCl buffer at 37°C for 20 mins. Fluorescence was measured using the i3x-Spectrophotometer (Molecular Devices, USA), at excitation and emission wavelengths of 495 and 515 nm respectively for DAF-FM and 360 and 449 respectively for Calcein Blue. After reading baseline fluorescence, 1 mM L-arginine was added to acquire fluorescence of live NO production. The fluorescence measurements were expressed as the change in fluorescence relative to the fluorescence measurement just before arginine addition. To account for the heterogeneity of the organoid cultures, final readouts were expressed as DAF-FM fluorescence/Calcein Blue AM fluorescence. Each condition was repeated with at least four technical replicates on the sample plate, and a total of at least four biological replicates per condition. Analysis was performed as previously described (*Ahmadi et al., 2017*). As shown in *Figure 6—figure supplement 3, a* standard curve was generated by increasing intracellular NO levels using a NO donor (Proli NONOate).

## Caco-2 cell culture and FLIPR based CFTR functional assay

Caco-2 cells were obtained from ATCC (ATCC HTB-37). Cells were cultured using Eagle's Minimum Essential Medium (EMEM) with 20% FBS (Fetal Bovine Serum), and 1% penicillin-streptomycin solution. Cells were grown at 37°C in presence of 5% $CO_2$. CFTR functional assay using the blue FLIPR® membrane potential dye was performed as previously described (*Ahmadi et al., 2017*; *Molinski et al., 2015*). Cells were plated on black clear bottom 96-well plates and assay was performed 1 week post-confluence. On the day of the assay cells were washed with PBS (Phosphate Buffered Solution) and FLIPR® blue membrane potential dye was dissolved in buffer (112.5 mM NMDG- Gluconate, 36.25 mM NaCl, 2.25 mM KGluconate, 0.75 mM KCl, 0.75 mM $CaCl_2$, 0.5 mM $MgCl_2$ and 10 mM HEPES, 300 mOsm, pH 7.35). Dye was used at a concentration of 0.5 mg/ml, and 100 µL of dye was added to each well. After 40 mins of dye loading, the cells were read using a fluorescence plate reader (Molecular Devices i3x) at an excitation wavelength of 530 nm and emission wavelength of 560 nm. After reading baseline for at least 5 mins, cells were treated with

pharmacological modulators (2.5 µL/well). Data were analyzed as previously described (*Ahmadi et al., 2017*), and statistics were performed using GraphPad Prism v6.01.

## Epithelial cell line transduction

The Caco-2 and CFBE41o⁻ bronchial cell lines were validated as expressing CFTR. The CFBE41o⁻ and Caco-2 cell lines were confirmed as mycoplasma negative. Caco-2 cells were cultured until 80% confluence. Cells were then transduced with lentivirus vector packaging *LV-SLC6A14-GFP* (C-terminal GFP) or *LV-GFP* at MOI 50 (Multiplicity of infection 50). Medium containing lentivirus particles was removed after 24 hr. Medium was changed every day for 2 days. Transduction was confirmed using western blotting.

## Western blotting

Cells from multi-well plates were lysed using the modified radioimmunoprecipitation assay buffer (50 mM Tris-HCl, 150 mM NaCl, 1 mM EDTA, pH 7.4, 0.1% (v/v) SDS, and 1% (v/v) Triton X-100) containing a protease inhibitor cocktail (Roche) for 5–10 mins. The lysates were then spun down at max speed (18,000 *g*) for 5 mins using a table-top centrifuge. The supernatant was then collected in a separate tube and laemmli sample buffer was added (1/5 dilution), and then sample was run on a SDS-PAGE gel. The protein from the gel was then transferred to a nitrocellulose membrane and blocked with 5% (w/v) milk. After blocking, the membrane was incubated with one of the following antibodies: human CFTR 596 (1:10000, UNC-CH and CFFT, UNC-CH University of North Carolina Chapel Hill, CFFT Cystic Fibrosis Foundation Therapeutics), human calnexin (1:10000, Sigma), human anti-β-actin antibodies (1:10000), GFP antibody (1:3000) or anti-FLAG antibody (1:1000, Sigma). After incubation with any of the above, the membrane was incubated with horseradish peroxidase conjugated secondary antibody raised in goat (against mouse or rabbit primary antibody, 1:2000 dilution), and after multiple washes chemi-luminescence signal was detected using the Li-Cor Odyssey Fc image acquisition system. The images were exported in tag image file format (TIFF), and analyzed using ImageJ 1.48 software (National Institutes of Health).

## Co-Immunoprecipitation

Baby hamster kidney (BHK) cells stably expressing wild-type or F508del-CFTR were transfected with vector expressing wild-type *SLC6A14*, bearing a C-terminal FLAG tag (pcDNA3.1) or control empty vector. Transfected F508del-CFTR BHK cells were incubated over night at 27°C. Cells were then lysed with ice cold lysis buffer (50 mM EDTA, 150 mM Tris base, 50 mM NaCl, 1% TritonX-100, pH 7.4) and incubated at 4°C overnight with monoclonal anti-FLAG antibody (Sigma-Aldrich, UT, USA). Lysate was incubated with protein A/G PLUS-Agarose beads (Santa Cruz Biotechnology, TX, USA) at 4°C for approximately 3 hr. Samples were then spun down and washed twice in RIPA buffer (50 mM Tris HCl, 150 mM NaCl, protease inhibitors pH 7.4), followed by two additional washed with lysis buffer. Immunoprecipitated samples were reduced using sample buffer (315 mM Tris base, 10% SDS, 125 mM DTT, 50% glycerol, 0.025% bromophenol blue dye, pH 6.8), 10 mM TCEP and DDT and titrated with Tris base. The final lysate containing immunoprecipitated samples were removed from the agarose beads through suction using a 27 ½ gauge needle. Samples were analyzed using western blotting, as described above.

## Cycloheximide and Brefeldin A chase assay

Baby hamster kidney (BHK) cells stably expressing wild-type or F508del-CFTR were transfected with vector expressing wild-type S*LC6A14* bearing a C-terminal FLAG tag (pcDNA3.1) or control empty vector. Transfected F508del-CFTR BHK cells were incubated over night at 27°C. Cells were then lysed with ice cold lysis buffer (50 mM EDTA, 150 mM Tris base, 50 mM NaCl, 1% TritonX-100, 0.1% SDS, pH 7.4). Temperature rescued cells were then treated with DMEM/F12 media containing a final concentration of 500 µg/mL of Cycloheximide and 10 µg/mL of Brefeldin A. Cells were lysed at different time points. Protein expression was determined through western blotting and analyzed with ImageJ 1.48 v.

## Statistics

One-way ANOVA with Tukey's multiple comparison test was performed on all data with more than two data-sets for comparison, and SD or SEM was calculated using data from biological replicates. For mice survival data analysis, Log Rank test was used. Unpaired two-tailed t-test was performed on data with two data-sets, and a paired t-test was performed where the data were paired from littermates. In vivo and in vitro experiments from each mouse was treated as a biological replicate. Technical replicates were defined as different replicates from sample obtained from the same mouse. $p < 0.05$ was considered statistically significant. Statistical analyses were performed using GraphPad Prism 6.01.

## Acknowledgements

We are thankful to the staff at the Laboratory Animal Services (LAS) at The Hospital for Sick Children for their help in maintaining the mouse colony, the Toronto Center for Phenogenomics (TCP) and Dr. Lauryl Nutter for help in generating the *Slc6a14* knock-out mice. We are grateful to CFFT (Cystic Fibrosis Foundation Therapeutics) for providing the CFTR modulators and to Dr. Bob Scholte, Erasmus Medical Center Rotterdam, Netherlands, for providing the FVB *Cftr*[F508del] mice. We are also grateful to Hayley Craig-Barnes and the Analytical Facility for Bioactive Molecules for help with Mass Spectrometry, to Reynaldo Interior (SPARC, Hospital for Sick Children) for his help with HPLC, to Alexandria Lew for her help with western blotting of Caco-2 cell lysates, to members of Dr. Nicola Jones' laboratory for their help with organoid culture, to the Imaging Facility at the Hospital for Sick Children for their technical help with imaging, and to Dr. Hartmut Grasemann, Dr. Tanja Gonska, Dr. Lisa Strug and Dr. Jaques Belik for their helpful discussions. CFBE41o⁻ cell line was a generous gift from Dr. Deiter Gruenert. We thank Jacqueline McCormack for critical reading of the manuscript. We are thankful to all members of Dr. Christine E Bear's laboratory and to all members of Dr. Johanna Rommens' laboratory for their helpful suggestions. These studies were supported by Operating Grants to CEB from the Canadian Institutes of Health Research (CIHR MOP-97954, CIHR GPG-102171) and Cystic Fibrosis Canada. SA was funded from operating grants from Dr. Christine E Bear, Dr. Johanna Rommens and by the Dr. Albert and Dorris Fields Graduate Scholarship. SX was funded from CIHR M CGS award and from operating grants from Dr. Christine E Bear.

## Additional information

### Funding

| Funder | Grant reference number | Author |
| --- | --- | --- |
| Canadian Institutes of Health Research | CIHR MOP-97954 | Fan Lin |
| Canadian Institutes of Health Research | CIHR GPG-102171 | Christine E Bear |
| Cystic Fibrosis Canada | CFC-1 | Christine E Bear |

The funders had no role in study design, data collection and interpretation, or the decision to submit the work for publication.

### Author contributions

Saumel Ahmadi, Conceptualization, Formal analysis, Investigation, Methodology, Writing—original draft; Sunny Xia, Yu-Sheng Wu, Formal analysis, Investigation, Visualization, Methodology, Writing—review and editing; Michelle Di Paola, Formal analysis, Investigation, Methodology, Writing—review and editing; Randolph Kissoon, Formal analysis, Investigation, Visualization, Methodology; Catherine Luk, Fan Lin, Investigation, Methodology; Kai Du, Investigation, Visualization, Methodology; Johanna Rommens, Conceptualization, Funding acquisition, Investigation; Christine E Bear, Conceptualization, Resources, Supervision, Funding acquisition, Writing—original draft, Project administration, Writing—review and editing

## Author ORCIDs

Saumel Ahmadi (iD) http://orcid.org/0000-0002-5800-6945
Sunny Xia (iD) http://orcid.org/0000-0002-3571-0462
Yu-Sheng Wu (iD) http://orcid.org/0000-0003-2882-9120
Michelle Di Paola (iD) http://orcid.org/0000-0003-1303-1407
Christine E Bear (iD) https://orcid.org/0000-0001-7063-3418

## Ethics

Animal experimentation: SickKids Animal Approval: #1000037433

## Decision letter and Author response

Decision letter https://doi.org/10.7554/eLife.37963.032
Author response https://doi.org/10.7554/eLife.37963.033

## Additional files

### Supplementary files

• Supplementary file 1. Serum amino-acid levels measured in 56 days old mice across four genotypes - Wt, CF ($Cftr^{F508del/F508del}$), $Slc6a14^{(-/y)}$ and double mutant ($Cftr^{F508del/F508del}$; $Slc6a14^{(-/y)}$). Mean and standard deviation (SD) are shown (n ≥ 4 for each genotype).
DOI: https://doi.org/10.7554/eLife.37963.028

• Supplementary file 2. *Slc6a14* deletion in CF mice maintains Mendelian inheritance. Table shows observed and expected distribution of mice with different genotypes.
DOI: https://doi.org/10.7554/eLife.37963.029

• Transparent reporting form
DOI: https://doi.org/10.7554/eLife.37963.030

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
