## [Decision Letter]

Thank you for submitting your article "SLC6A14, an amino acid transporter, modifies the primary CF defect in fluid secretion" for consideration by *eLife*. Your article has been reviewed by three peer reviewers, including Ralph DeBerardinis as Reviewer #1, and the evaluation has been overseen by Richard Aldrich as the Senior Editor. The following individuals involved in review of your submission have agreed to reveal their identity: Carol Bertrand (Reviewer #2); Lane Clarke (Reviewer #3).

The reviewers have discussed the reviews with one another and the Reviewing Editor has drafted this decision to help you prepare a revised submission.

Summary:

This paper explores the role of the amino acid transporter SLC6A14 in modifying the intestinal phenotype of cystic fibrosis caused by homozygosity of the CFTR delta F508 mutation. The authors find that loss of SLC6A14 reduced arginine abundance in colonic tissue and, when combined with delta F508 homozygosity, led to reduced survival after weaning. SLC6A14 contributes to an arginine-dependent phenotype of CFTR-mediated fluid secretion. The authors provide evidence that this mechanism involves nitric oxide (NO) production by iNOS, which requires arginine as a substrate. NO stimulates protein kinase G-mediated stimulation of CFTR activity and thereby enhances mutant CFTR's secretory activity.

Essential revisions:

Overall, all three reviewers were enthusiastic about the findings of the paper. Three substantive comments are listed below. In addition, one reviewer suggested that confirmation of some of these observations in primary human colonic tissue would strengthen the paper.

1) In Figure 8, the authors show that a permeable form of cGMP can enhance FSK-stimulated CFTR activity. This instructive experiment should be complemented by an experiment that more fully tests the model. In F508del/SLC6A14 WT organoids, do NO donors, in the absence of Arg, also stimulate CFTR activity, and can this be blocked by PKG inhibitors?

2) SLC6A14 is a Na- and Cl-dependent transporter, but the impact of luminal electrolyte concentration on amino acid uptake was not tested or discussed. This is important in the context of CFTR, as mutation-dependent changes in the near membrane chloride concentration may occur, and evidence suggests that removal of chloride substantially impairs amino acid uptake (i.e., Anderson et al., 2008). The majority of functional assays were performed in the presence of 140 mM NaCl, which likely results in maximum transport. Any available experiments or, at a minimum, citation of literature regarding the ion dependence should be presented in the context of how aberrant Cl^-^ secretion might impact transport.

3) A technical concern is raised with the ex vivo loop assay for assessing intestinal arginine uptake (subsection “Ex vivo closed loop amino acid uptake assay”: Figure 3E and 3F). From the appearance of colonic sections shown in Figure 3A and 3B, distinct morphological differences are apparent in the volume of mucosa vs. the combined volume of submucosal and smooth muscle in the F508del ± *Slc6a14* knockout as compared to WT genotypes. Since the assay reports arginine CPM/mg/ml protein from protein isolates of whole intestine, the mucosal protein as a fraction of total protein is likely less in the F508del ± *Slc6a14* knockout as compared to WT genotypes, thereby accentuating the purported decrease in uptake in the F508del CFTR cohorts. Since the authors have tissue volumes of the different genotypes (measured by 3DHISTECH), a calculation of these differences would indicate whether there is potential for a significant error in the arginine uptake data. If so, arginine uptake needs a different normalization standard or correction by mathematical means.

---

## [Author Response]

Essential revisions:Overall, all three reviewers were enthusiastic about the findings of the paper. Three substantive comments are listed below. In addition, one reviewer suggested that confirmation of some of these observations in primary human colonic tissue would strengthen the paper.1) In Figure 8, the authors show that a permeable form of cGMP can enhance FSK-stimulated CFTR activity. This instructive experiment should be complemented by an experiment that more fully tests the model. In F508del/SLC6A14 WT organoids, do NO donors, in the absence of Arg, also stimulate CFTR activity, and can this be blocked by PKG inhibitors?

We did our best to address this suggestion in our revision with new experimental data using F508del mouse intestinal organoids to complement mechanistic studies in the human colonic cell line: CaCo-2.

We conducted new studies in organoids derived from a male F508del/SLC6A14 Wt mouse. In triplicate organoid platings, we showed that the NO donor: GSNO stimulated residual F508del channel function in the absence of arginine (subsection “Loss of *Slc6a14* and arginine-mediated nitric oxide generation contributes to worsening of defective epithelial fluid secretion”, fifth paragraph, plus Figure 6—figure supplement 2).

We showed the role of PKG in this NO effect in split open organoids prepared from the human colonic, CaCo-2 cell line bearing Wt-CFTR (Figure 7). The use of this cell line enabled a mid-throughput study comparing multiple interventions: an inhibitor of iNOS, two different NO donors and modulation of GSK using two chemically distinct small molecules. In our revision, we highlighted the importance of these detailed, mechanistic studies in CaCo-2 cells for supporting relevance in humans and validating the steps in our proposed signaling pathway (subsection “Loss of *Slc6a14* and arginine-mediated nitric oxide generation contributes to worsening of defective epithelial fluid secretion”, seventh paragraph).

2) SLC6A14 is a Na- and Cl-dependent transporter, but the impact of luminal electrolyte concentration on amino acid uptake was not tested or discussed. This is important in the context of CFTR, as mutation-dependent changes in the near membrane chloride concentration may occur, and evidence suggests that removal of chloride substantially impairs amino acid uptake (i.e., Anderson et al., 2008). The majority of functional assays were performed in the presence of 140 mM NaCl, which likely results in maximum transport. Any available experiments or, at a minimum, citation of literature regarding the ion dependence should be presented in the context of how aberrant Cl^-^ secretion might impact transport.

As suggested by our reviewer, we revised the text in the Results section to highlight the requirement of SLC6A14 for NaCl. According to the literature, the K_M_ (Na^+^) is 10-20 mM and the K_M_ (Cl^-^) is 1-2 mM for SLC6A14 (Karunakaran et al., 2011). Electrolyte concentrations in the stool have been estimated at 5-38 mM (Comparison of the composition of faecal fluid in Crohn's disease and ulcerative colitis Gut, 1982,23,326-332) and not expected to be limiting to the activity of the SLC6A14 transporter. In our functional studies of the transporter we exceeded these stool electrolyte concentrations in both the closed loop assay (140 mM NaCl) and in the ACC assay of split open organoids (38 mM NaCl). Therefore, we have likely modeled maximal transport efficiency in both systems. These considerations have been included in the revised version of our manuscript (subsection “Ex vivo closed loop amino acid uptake assay”, first paragraph and subsection “Apical Chloride Conductance (ACC) Assay for CFTR in “split-open” organoids”).

3) A technical concern is raised with the ex vivo loop assay for assessing intestinal arginine uptake (subsection “Ex vivo closed loop amino acid uptake assay”: Figure 3E and 3F). From the appearance of colonic sections shown in Figure 3A and 3B, distinct morphological differences are apparent in the volume of mucosa vs. the combined volume of submucosal and smooth muscle in the F508del ± Slc6a14 knockout as compared to WT genotypes. Since the assay reports arginine CPM/mg/ml protein from protein isolates of whole intestine, the mucosal protein as a fraction of total protein is likely less in the F508del ± Slc6a14 knockout as compared to WT genotypes, thereby accentuating the purported decrease in uptake in the F508del CFTR cohorts. Since the authors have tissue volumes of the different genotypes (measured by 3DHISTECH), a calculation of these differences would indicate whether there is potential for a significant error in the arginine uptake data. If so, arginine uptake needs a different normalization standard or correction by mathematical means.

The reviewer raises an interesting point however, we compared arginine uptake in intestinal loops obtained from Wt versus SLC6A14 knockout mice where there was no significant difference in weight or morphology. In the revised version we include a statement regarding the lack of effect of SLC6A14 knockout on morphology (Discussion, third paragraph). Our findings are substantiated by recent phenotypic data generated by the NorComm2 Phenotyping consortium through which these mice were created (reference in the aforementioned paragraph). Therefore, the decrease in arginine uptake associated with knockout of SLC6A14 is not reflecting a change in tissue morphology.